# Towards Identifiable Unsupervised Domain Translation: A Diversified Distribution Matching Approach

**Sagar Shrestha & Xiao Fu** [*]
School of Electrical Engineering and Computer Science
Oregon State University
Corvallis, OR 97331, USA
{shressag,xiao.fu}@oregonstate.edu

## Abstract

Unsupervised domain translation (UDT) aims to find functions that convert samples from one domain (e.g., sketches) to another domain (e.g., photos) without changing the high-level semantic meaning (also referred to as "content"). The translation functions are often sought by probability distribution matching of the transformed source domain and target domain. CycleGAN stands as arguably the most representative approach among this line of work. However, it was noticed in the literature that CycleGAN and variants could fail to identify the desired translation functions and produce content-misaligned translations. This limitation arises due to the presence of multiple translation functions—referred to as "measure-preserving automorphism" (MPA)—in the solution space of the learning criteria. Despite awareness of such identifiability issues, solutions have remained elusive. This study delves into the core identifiability inquiry and introduces an MPA elimination theory. Our analysis shows that MPA is unlikely to exist, if multiple pairs of diverse cross-domain conditional distributions are matched by the learning function. Our theory leads to a UDT learner using distribution matching over auxiliary variable-induced subsets of the domains—other than over the entire data domains as in the classical approaches. The proposed framework is the first to rigorously establish translation identifiability under reasonable UDT settings, to our best knowledge. Experiments corroborate with our theoretical claims.

## 1 Introduction

Domain translation (DT) aims to convert data samples from one feature domain to another, while keeping the key content information. DT naturally arises in many applications, e.g., transfer learning (Zhuang et al., 2020), domain adaptation (Ganin et al., 2016; Courty et al., 2017), and cross-domain retrieval (Huang et al., 2015). Among them, a premier application is image-to-image (I2I) translation (e.g., profile photo to cartonized emoji and satellite images to street map plots (Isola et al., 2017)). *Supervised* domain translation (SDT) relies on paired data from the source and target domains. There, the translation functions are learned via matching the sample pairs.

Nonetheless, paired data are not always available. In *unsupervised domain translation* (UDT), the arguably most widely adopted idea is to find neural transformation functions that perform probability distribution matching of the domains. The idea emerged in the literature in early works, e.g., (Liu & Tuzel, 2016; Taigman et al., 2017; Kim et al., 2017). High-resolution image translation using distribution matching was later realized by the seminal work, namely, CycleGAN (Zhu et al., 2017). CycleGAN learns a pair of transformations that are inverse of each other. One of transformations maps the source domain to match the distribution of the target domain, and the other transformation does the opposite. The distribution matching part is realized by the generative adversarial network (GAN) (Goodfellow et al., 2014). Using GAN-based distribution matching for UDT has attracted much attention—many follow-up works emerged; see the survey (Pang et al., 2021).

---

[*] Source code is available at https://github.com/XiaoFuLab/Identifiable-UDT.git

**Challenge - Lack of Translation Identifiability.** While UDT approaches have demonstrated significant empirical success, the theoretical question of translation identifiability has received relatively limited attention. Recent works (Galanti et al., 2018b;a; Moriakov et al., 2020; Galanti et al., 2021) pointed out failure cases of CycleGAN (e.g., content-misaligned translations like those in Fig. 1) largely attribute to the lack of translation identifiability. That is, translation functions in the solution space of CycleGAN (or any distribution matching-based learners) is non-unique, due to the existence of *measure-preserving automorphism* (MPA) (Moriakov et al., 2020) (the same concept was called *density-preserving mappings* in (Galanti et al., 2018b;a)). MPA can "swap" the cross-domain sample correspondences without changing the data distribution—which is likely the main source of producing content misaligned samples after translation as seen in Fig. 1. Many efforts were made to empirically enhance the performance of UDT, via implicitly or explicitly promoting solution uniqueness of their loss functions (Liu et al., 2017; Courty et al., 2017; Xu et al., 2022; Yang et al., 2023). A number of notable works approached the identifiability/uniqueness challenge by assuming that the desired translation functions have simple (e.g., linear (Gulrajani & Hashimoto, 2022)) or specific structures (de Bézenac et al., 2021). However, translation identifiability without using such restrictive structural assumptions have remained elusive.

**Contributions.** In this work, we revisit distribution matching-based UDT. Our contribution lies in both identifiability theory and implementation:

● **Theory Development: Establishing Translation Identifiability.** We delve into the core theoretical challenge regarding identifiability of the translation functions. As mentioned, the solution space of existing distribution matching criteria could be easily affected by MPA. However, our analysis shows that the chance of having MPA decreases quickly when the translation function aligns more than one pair of diverse distributions. This insight allows us to come up a sufficient condition, namely, the *sufficiently diverse condition* (SDC), to establish translation identifiability of UDT. To our best knowledge, our result stands as the first UDT identifiability theory without using simplified structural assumptions.

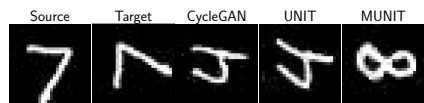

Figure 1: Lack of translation identifiability often leads to *content misalignment* in distribution matching based UDT methods, e.g., CycleGAN (Zhu et al., 2017), MUNIT(Huang et al., 2018), and UNIT (Liu et al., 2017). Source domain: MNIST Digits. Target Domain: Rotated Display of MNIST.

● **Simple Implementation via Auxiliary Variables.** Our theoretical revelation naturally gives rise to a novel UDT learning criterion. This criterion aligns multiple pairs of conditional distributions across the source and target domains. We define these conditional distributions over (overlapping) sub-domaions of the source/target domains using auxiliary variables. We demonstrate that in practical applications such as unpaired I2I translation, obtaining these sub-domains can be a straightforward task, e.g., through available side information or querying the foundation models like CLIP (Radford et al., 2021). Consequently, our identification theory can be readily put into practice.

**Notation.** The full list of notations is in the supplementary material. Notably, we use $\mathbb{P}_{\boldsymbol{x}}$ and $\mathbb{P}_{\boldsymbol{x}|u}$ to denote the *probability measures* of $\boldsymbol{x}$ and $\boldsymbol{x}$ conditioned on $u$, respectively. We denote the corresponding *probability density function* (PDF) of $\boldsymbol{x}$ by $p(\boldsymbol{x})$. For a measurable function $\boldsymbol{f} : \mathcal{X} \to \mathcal{Y}$ and a distribution $\mathbb{P}_{\boldsymbol{x}}$ defined over space $\mathcal{X}$, the notation $\boldsymbol{f}_{\#\mathbb{P}_{\boldsymbol{x}}}$ denotes the *push-forward measure*; that is, for any measurable set $\mathcal{A} \subseteq \mathcal{Y}$, $\boldsymbol{f}_{\#\mathbb{P}_{\boldsymbol{x}}}[\mathcal{A}] = \mathbb{P}_{\boldsymbol{x}}[\boldsymbol{f}^{\mathrm{preimg}}(\mathcal{A})]$, where $\boldsymbol{f}^{\mathrm{preimg}}(\mathcal{A}) = \{\boldsymbol{x} \in \mathcal{X} \mid \boldsymbol{f}(\boldsymbol{x}) \in \mathcal{A}\}$. Simply speaking, $\boldsymbol{f}_{\#\mathbb{P}_{\boldsymbol{x}}}$ denotes the distribution of $\boldsymbol{f}(\boldsymbol{x})$ where $\boldsymbol{x} \sim \mathbb{P}_{\boldsymbol{x}}$. The notation $\boldsymbol{f}_{\#\mathbb{P}_{\boldsymbol{x}}} = \mathbb{P}_{\boldsymbol{y}}$ means that the PDFs of $\boldsymbol{f}(\boldsymbol{x})$ and $\boldsymbol{y}$ are identical *almost everywhere* (a.e.).

## 2 PRELIMINARIES

Considers two data domains (e.g., photos and sketches). The samples from the two domains are represented by $\boldsymbol{x} \in \mathcal{X} \subseteq \mathbb{R}^{D_x}$ and $\boldsymbol{y} \in \mathcal{Y} \subseteq \mathbb{R}^{D_y}$. We make the following assumption:

**Assumption 1.** *For every $\boldsymbol{x} \in \mathcal{X}$, it has a corresponding $\boldsymbol{y} \in \mathcal{Y}$, and vice versa. In addition, there exist deterministic continuous functions $\boldsymbol{f}^\star : \mathcal{Y} \to \mathcal{X}$ and $\boldsymbol{g}^\star : \mathcal{X} \to \mathcal{Y}$ that link the corresponding pairs; i.e.,*

$$\boldsymbol{f}^\star(\boldsymbol{y}) = \boldsymbol{x}, \quad \boldsymbol{g}^\star(\boldsymbol{x}) = \boldsymbol{y}, \quad \forall \text{ corresponding pair } (\boldsymbol{x}, \boldsymbol{y}). \tag{1}$$

In the context of domain translation, a linked $(\boldsymbol{x}, \boldsymbol{y})$ pair can be regarded as cross-domain data samples that represent the same "content", and the translation functions $(\boldsymbol{f}^\star, \boldsymbol{g}^\star)$ are responsible for changing their "appearances/styles". The term "content" refers to the semantic information to be kept across domains after translation. In Fig. 1, the content is the identity of the digit (other than writing style or the rotation); in Fig. 4 of Sec. 3, the content can be understood as the shared characteristics of the person in both the cartoon and the photo domains, which can collectively identify the person.

Note that in the above setting, the goal is to find *two* ground-truth translation functions where one function's source is the other's target. Hence, both $\mathcal{X}$ and $\mathcal{Y}$ can serve as the source/target domains. In addition, the above also implies $\boldsymbol{f}^\star = (\boldsymbol{g}^\star)^{-1}$, i.e., the ground-truth translation functions are invertible. Under this setting, if one can identify $\boldsymbol{g}^\star$ and $\boldsymbol{f}^\star$, then the samples in one domain can be translated to the other domain—while not changing the content. Note that Assumption 1 means that there is one-to-one correspondence between samples in the two domains, which can be a somewhat stringent condition in some cases. However, as we will explain in detail later, many UDT works, e.g., CycleGAN (Zhu et al., 2017) and variants (Liu et al., 2017; Kim et al., 2017; Choi et al., 2018; Park et al., 2020), essentially used the model in Assumption 1 to attain quite interesting empirical results. This makes it a useful model and intrigues us to understand its underlying properties.

**Supervised Domain Translation (SDT).** In SDT, the corresponding pairs $(\boldsymbol{x}, \boldsymbol{y})$ are assumed to be aligned *a priori*. Then, learning a translation function is essentially a regression problem—e.g., via finding $\boldsymbol{g}$ (or $\boldsymbol{f}$) such that $D(\boldsymbol{g}(\boldsymbol{x})\|\boldsymbol{y})$ (or $D(\boldsymbol{f}(\boldsymbol{y})\|\boldsymbol{x})$) is minimized over all given pairs, where $D(\cdot\|\cdot)$ is a certain "distance" measure; see, e.g., (Isola et al., 2017; Wang et al., 2018).

**Unsupervised Domain Translation (UDT).** In UDT, samples from the two domains are acquired separately without alignment. Hence, sample-level matching as often done in SDT is not viable. Instead, UDT is often formulated as a probability distribution matching problem (see, e.g., (Zhu et al., 2017; Taigman et al., 2017; Kim et al., 2020; Park et al., 2020))—as distribution matching can be attained without using sample-level correspondences. Assume that $\boldsymbol{x}$ and $\boldsymbol{y}$ are the random vectors that represent the data from the $\mathcal{X}$-domain and the $\mathcal{Y}$-domain, respectively. Then, the desired $\boldsymbol{f}^\star$ and $\boldsymbol{g}^\star$ are sought via finding $\boldsymbol{f}$ and $\boldsymbol{g}$ such that

$$\mathbb{P}_{\boldsymbol{y}} = \boldsymbol{g}_{\#\mathbb{P}_{\boldsymbol{x}}} \quad \text{and} \quad \mathbb{P}_{\boldsymbol{x}} = \boldsymbol{f}_{\#\mathbb{P}_{\boldsymbol{y}}}. \tag{2}$$

The hope is that distribution matching can work as a surrogate of sample-level matching as in SDT. The arguably most representative work in UDT is CycleGAN (Zhu et al., 2017). The CycleGAN loss function is as follows:

$$\min_{\boldsymbol{f}, \boldsymbol{g}} \max_{\boldsymbol{d}_x, \boldsymbol{d}_y} \mathcal{L}_{\mathrm{GAN}}(\boldsymbol{g}, \boldsymbol{d}_y, \boldsymbol{x}, \boldsymbol{y}) + \mathcal{L}_{\mathrm{GAN}}(\boldsymbol{f}, \boldsymbol{d}_x, \boldsymbol{x}, \boldsymbol{y}) + \lambda \mathcal{L}_{\mathrm{cyc}}(\boldsymbol{g}, \boldsymbol{f}), \tag{3}$$

where $\boldsymbol{d}_x$ and $\boldsymbol{d}_y$ represent two discriminators in domains $\mathcal{X}$ and $\mathcal{Y}$, respectively,

$$\mathcal{L}_{\mathrm{GAN}}(\boldsymbol{g}, \boldsymbol{d}_y, \boldsymbol{x}, \boldsymbol{y}) = \mathbb{E}_{\boldsymbol{y} \sim \mathbb{P}_{\boldsymbol{y}}}[\log \boldsymbol{d}_y(\boldsymbol{y})] + \mathbb{E}_{\boldsymbol{x} \sim \mathbb{P}_{\boldsymbol{x}}}[\log(1 - \boldsymbol{d}_y(\boldsymbol{g}(\boldsymbol{x})))], \tag{4}$$

$\mathcal{L}_{\mathrm{GAN}}(\boldsymbol{f}, \boldsymbol{d}_x, \boldsymbol{x}, \boldsymbol{y})$ is defined in the same way, and the cycle-consistency term is defined as

$$\mathcal{L}_{\mathrm{cyc}}(\boldsymbol{g}, \boldsymbol{f}) = \mathbb{E}_{\boldsymbol{x} \sim \mathbb{P}_{\boldsymbol{x}}}[\|\boldsymbol{f}(\boldsymbol{g}(\boldsymbol{x})) - \boldsymbol{x}\|_1] + \mathbb{E}_{\boldsymbol{y} \sim \mathbb{P}_{\boldsymbol{y}}}[\|\boldsymbol{g}(\boldsymbol{f}(\boldsymbol{y})) - \boldsymbol{y}\|_1]. \tag{5}$$

The minimax optimization of the $\mathcal{L}_{\mathrm{GAN}}$ terms enforces $\boldsymbol{g}_{\#\mathbb{P}_{\boldsymbol{x}}} = \mathbb{P}_{\boldsymbol{y}}$ and $\boldsymbol{f}_{\#\mathbb{P}_{\boldsymbol{y}}} = \mathbb{P}_{\boldsymbol{x}}$. The $\mathcal{L}_{\mathrm{cyc}}$ term encourages $\boldsymbol{f} = \boldsymbol{g}^{-1}$. CycleGAN showed the power of distribution matching in UDT and has triggered a lot of interests in I2I translation. Many variants of CycleGAN were also proposed to improve the performance; see the survey (Pang et al., 2021).

**Lack of Translation Identifiability, MPA and Content Misalignment.** Many works have noticed that distribution matching-type learning criterion may suffer from the lack of translation identifiability (Liu et al., 2017; Moriakov et al., 2020; Galanti et al., 2018b; 2021; Xu et al., 2022); i.e., the solution space of these criteria could have multiple solutions, and thus lack the ability to recover the ground-truth $\boldsymbol{g}^\star$ and $\boldsymbol{f}^\star$. The lack of identifiability often leads to issues such as content misalignment as we saw in Fig. 1. To understand the identifiability challenge, let us formally define identifiability of any bi-directional UDT learning criterion:

**Definition 1.** *(Identifiability) Under the setting of Assumption 1, assume that $(\widehat{\boldsymbol{f}}, \widehat{\boldsymbol{g}})$ is any optimal solution of a UDT learning criterion. Then, identifiability of $(\boldsymbol{f}^\star, \boldsymbol{g}^\star)$ holds under the UDT learning criterion if and only if $\widehat{\boldsymbol{f}} = \boldsymbol{f}^\star$ and $\widehat{\boldsymbol{g}} = \boldsymbol{g}^\star$ a.e.*

Notice that we used the *optimal solution* in the definition. This is because identifiability is a characterization of the "kernel space" (which contains all the zero-loss solutions) of a learning criterion (Moriakov et al., 2020; Fu et al., 2019). In other words, when a UDT criterion admits translation identifiability, it indicates that the criterion provides a valid objective for the learning task—but identifiability is not related to the optimization procedure. We will also use the following:

**Definition 2.** *(MPA) A measure-preserving automorphism (MPA) of $\mathbb{P}_{\boldsymbol{x}}$ is a continuous function $\boldsymbol{h} : \mathcal{X} \to \mathcal{X}$ such that $\mathbb{P}_{\boldsymbol{x}} = \boldsymbol{h}_{\#\mathbb{P}_{\boldsymbol{x}}}$.*

Simply speaking, MPA defined in this work is the continuous transformation $\boldsymbol{h}(\boldsymbol{x})$ whose output has the same PDF as $p(\boldsymbol{x})$. Take the one-dimensional Gaussian distribution $x \sim \mathcal{N}(\mu, \sigma^2)$ as an example. The MPA of $\mathcal{N}(\mu, \sigma^2)$ is $h(x) = -x + 2\mu$. A recent work (Moriakov et al., 2020) suggested that non-identifiability of the desired translation functions by CycleGAN is caused by the existence of MPA. Their finding can be summarized in the following Fact:

**Fact 1.** *If MPA of $\mathbb{P}_{\boldsymbol{x}}$ or $\mathbb{P}_{\boldsymbol{y}}$ exists, then CycleGAN and any criterion using distribution matching in (2) do not have identifiability of $\boldsymbol{f}^\star$ and $\boldsymbol{g}^\star$.*

*Proof*: It is straightforward to see that $\mathbb{P}_{\boldsymbol{y}} = \boldsymbol{g}^\star_{\#\mathbb{P}_{\boldsymbol{x}}}$ and $\mathbb{P}_{\boldsymbol{x}} = \boldsymbol{f}^\star_{\#\mathbb{P}_{\boldsymbol{y}}}$. In addition, $\boldsymbol{f}^\star$ and $\boldsymbol{g}^\star$ are invertible. Hence, the ground truth $(\boldsymbol{f}^\star, \boldsymbol{g}^\star)$ is an optimal solution of CycleGAN that makes the loss in (3) equal to zero. However, due to the existence MPA, one can see that $\widehat{\boldsymbol{f}} = \boldsymbol{h} \circ \boldsymbol{f}^\star$ can also attain $\mathbb{P}_{\boldsymbol{x}} = \widehat{\boldsymbol{f}}_{\#\mathbb{P}_{\boldsymbol{y}}}$. This is because we have $\widehat{\boldsymbol{f}}_{\#\mathbb{P}_{\boldsymbol{y}}} = \boldsymbol{h} \circ \boldsymbol{f}^\star_{\#\mathbb{P}_{\boldsymbol{y}}} = \boldsymbol{h}_{\#\mathbb{P}_{\boldsymbol{x}}} = \mathbb{P}_{\boldsymbol{x}}$.

Plus, as $\boldsymbol{h} \circ \boldsymbol{f}^\star$ is still invertible, $\widehat{\boldsymbol{f}}$ still makes the cycle-consistency loss zero. Hence, the solution of CycleGAN is not unique and this loses identifiability of the ground truth translation functions. $\square$

The existence of MPA in the solution space of the UDT learning losses may be detrimental in terms of avoiding content misalignment. To see this, consider the example in Fig. 2. There, $\mathbb{P}_x = \mathcal{N}(\mu, \sigma^2)$ and $\boldsymbol{h}(x) = -x + 2\mu$ is an MPA of $\mathbb{P}_x$, as mentioned. Note that $\widehat{\boldsymbol{f}} = \boldsymbol{h} \circ \boldsymbol{f}^\star$ can be an optimal solution found by CycleGAN. However, such an $\widehat{\boldsymbol{f}}$ can cause misalignment. To explain, assume $x = a$ and $y = b$ are associated with the same entity, which means that $a = \boldsymbol{f}^\star(b)$ represents the ground-truth alignment and translation. However, as $p(-a + 2\mu) = p(\boldsymbol{h}(a)) = p(\boldsymbol{h} \circ \boldsymbol{f}^\star(b)) = p(\widehat{\boldsymbol{f}}(b))$, the learned function $\widehat{\boldsymbol{f}}$ wrongly translates $y = b$ to $x = -a + 2\mu$.

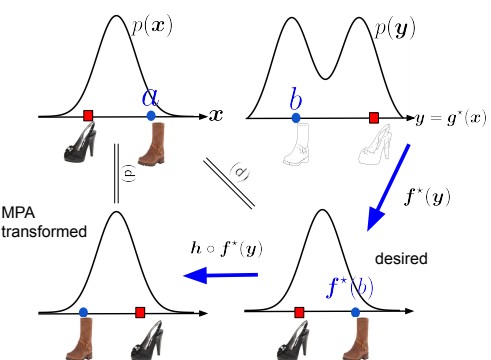

Figure 2: Illustration of of the lack of identifiability and MPA-induced content misalignment; "$\overset{(d)}{=\!=\!=}$" means distribution matching.

Our Gaussian example seems to be special as it has symmetry about its mean. However, the existence of MPA is not unusual. To see this, we show the following result:

**Proposition 1.** *Suppose that $\mathbb{P}_{\boldsymbol{x}}$ admits a continuous PDF, $p(\boldsymbol{x})$ and $p(\boldsymbol{x}) > 0, \forall \boldsymbol{x} \in \mathcal{X}$. Assume that $\mathcal{X}$ is simply connected. Then, there exists a continuous non-trivial (non-identity) $\boldsymbol{h}(\cdot)$ such that $\boldsymbol{h}_{\#\mathbb{P}_{\boldsymbol{x}}} = \mathbb{P}_{\boldsymbol{x}}$.*

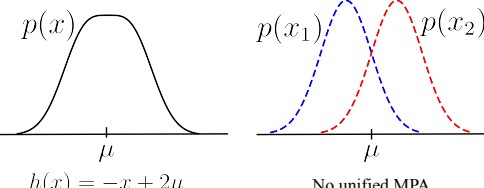

Figure 3: A unified MPA is harder to exist for a group of distributions.

Note that there are similar results in (Moriakov et al., 2020) regarding the existence of MPA, but more assumptions were made in their proof. The universal existence of MPA attests to the challenging nature of establishing translation identfiability in UDT.

## 3 IDENTIFIABLE UDT VIA DIVERSIFIED DISTRIBUTION MATCHING

**Intuition - Exploiting Diversity of Distributions.** Our idea starts with the following observation: If two distributions have different PDFs, a shared MPA is unlikely to exist. Fig. 3 illustrates the

intuition. Consider two Gaussian distributions $x_1 \sim \mathcal{N}(\mu_1, 1)$ and $x_2 \sim \mathcal{N}(\mu_2, 1)$ with $\mu_1 \neq \mu_2$. For each of them, $h(x) = -x + 2\mu_i$ for $i = 1, 2$ is an MPA. However, there is not a function that can serve as a unified MPA to attain $h_{\#\mathbb{P}_{x_1}} = \mathbb{P}_{x_1}$ & $h_{\#\mathbb{P}_{x_2}} = \mathbb{P}_{x_2}$ simultaneously. Intuitively, the diversity of the PDFs of $x_1$ and $x_2$ has made finding a unified MPA $h(\cdot)$ difficult. This suggests that instead of matching the distributions of $x$ and $f(y)$ and those of $y$ and $g(x)$, it may be beneficial to match the distributions of more variable pairs whose probability measures are diverse.

**Auxiliary Variable-Assisted Distribution Diversification.** In applications, the corresponding samples $x, y$ often share some aspects/traits. For example, in Fig. 4, the corresponding $x$ and $y$ both have dark hair or the same gender. If we model a collection of such traits as different realizations of discrete random variable $u$, the alphabet of $u$, denoted as $\{u_1, \ldots, u_I\}$ represents these traits. We should emphasize that the traits is a result of the desired content invariance across domains, but need not to represent the whole content.

To proceed, we observe that the conditional distributions $\mathbb{P}_{x|u=u_i}$ and $\mathbb{P}_{y|u=u_i}$ satisfy $\mathbb{P}_{x|u=u_i} = f^\star_{\#\mathbb{P}_{y|u=u_i}}$, $\mathbb{P}_{y|u=u_i} = g^\star_{\#\mathbb{P}_{x|u=u_i}}$, $\forall i$. The above holds since $x$ and $y$ have a deterministic relation and because the trait $u_i$ is shared by the content-aligned pairs $(x, y)$.

In practice, $u$ can take various forms. In I2I translation, one may use image categories or labels, if available, to serve as $u$. Note that knowing the image categories does *not* mean the samples from the two domains are aligned, as each category could contain a large amount of samples. In addition, one can use sample attributes (such as hair color, gender as in Fig. 4) to serve as $u$, if these attributes are not meant to be changed in the considered translation tasks. If not immediately available, these attributes can be annotated by open-sourced AI models, e.g., CLIP (Radford et al., 2021); see detailed implementation in the supplementary material. A similar idea of using CLIP to acquire auxiliary information was explored in (Gabbay et al., 2021).

By Proposition 1, it is almost certain that $\mathbb{P}_{x|u=u_i}$ has an MPA $h_i$ for all $i \in [I]$. However, it is likely that $h_i \neq h_j$ if $\mathbb{P}_{x|u=u_i}$ and $\mathbb{P}_{x|u=u_j}$ are sufficiently different. As a consequence, similar to what we saw in Fig. 3, if one looks for $f$ that does simultaneous matching of

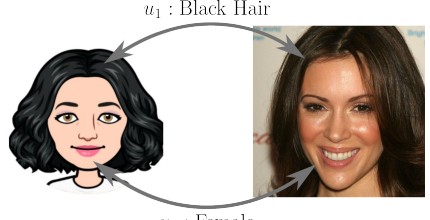

$$\mathbb{P}_{x|u=u_i} = f_{\#\mathbb{P}_{y|u=u_i}}, \ \forall i \in [I], \tag{6}$$

it is more possible that $f = f^\star$ instead of having other solutions—this leads to identfiiability of $f^\star$.

Figure 4: Examples of $u_i$.

**Proposed Loss Function.** We propose to match multiple distribution pairs $(\mathbb{P}_{x|u_i}, f_{\#\mathbb{P}_{y|u_i}})$ (as well as $(\mathbb{P}_{y|u_i}, g_{\#\mathbb{P}_{x|u_i}})$) for $i = 1, \ldots, I$. For each pair, we use discriminator $d_x^{(i)} : \mathcal{X} \to [0, 1]$ (and $d_y^{(i)} : \mathcal{Y} \to [0, 1]$ in reverse direction). Then, our loss function is as follows:

$$\min_{f, g} \max_{\{d_x^{(i)}, d_y^{(i)}\}} \sum_{i=1}^{I} \left( \mathcal{L}_{\text{GAN}}(g, d_y^{(i)}, x, y) + \mathcal{L}_{\text{GAN}}(f, d_x^{(i)}, x, y) \right) + \lambda \mathcal{L}_{\text{cyc}}(g, f), \tag{7}$$

where we have

$$\mathcal{L}_{\text{GAN}}\left(g, d_y^{(i)}, x, y\right) = \Pr(u = u_i) \left( \mathbb{E}_{y \sim \mathbb{P}_{y|u_i}} \left[ \log d_y^{(i)}(y) \right] + \mathbb{E}_{x \sim \mathbb{P}_{x|u_i}} \left[ \log \left( 1 - d_y^{(i)}(g(x)) \right) \right] \right).$$

Note that $x \sim \mathbb{P}_{x|u_i}$ represents samples that share the same characteristic defined by $u_i$ (e.g., hair color, eye color, gender). This means that the loss function matches a suite of distributions defined over (potentially overlapping) subdomains over the entire domain $\mathcal{X}$ and $\mathcal{Y}$. We should emphasize that the auxiliary variable is only needed in the training stage, but not the testing stage.

We call the proposed method *diversified distribution matching for unsupervised domain translation* (DIMENSION) [1]. The following lemma shows that DIMENSION exactly realizes our idea in (6):

**Lemma 1.** *Assume that an optimal solution of (7) is $(\widehat{f}, \widehat{g}, \{\widehat{d}_x^{(i)}, \widehat{d}_y^{(i)}\})$. Then, under Assumption 1, we have $\mathbb{P}_{x|u=u_i} = \widehat{f}_{\#\mathbb{P}_{y|u=u_i}}$, $\mathbb{P}_{y|u=u_i} = \widehat{g}_{\#\mathbb{P}_{x|u=u_i}}$, $\forall i \in [I]$, and $\widehat{f} = \widehat{g}^{-1}$, a.e.*

---

[1]Note that we still use the term "unsupervised" despite the need of auxiliary information—as no paired samples are required. We avoided using "semi-supervised" or "weakly supervised" as these are often reserved for methods using some paired samples; see, e.g., (Wang et al., 2020; Mustafa & Mantiuk, 2020).

**Identifiability Characterization.** Lemma 1 means that solving the `DIMENSION` loss leads to conditional distribution matching as we hoped for in (6). Hower, it does not guarantee that $(\widehat{\boldsymbol{f}}, \widehat{\boldsymbol{g}})$ found by `DIMENSION` satisfies $\widehat{\boldsymbol{f}} = \boldsymbol{f}^{\star}$ and $\widehat{\boldsymbol{g}} = \boldsymbol{g}^{\star}$. Towards establishing *identifiability* of the ground-truth translation functions via `DIMENSION`, we will use the following definition:

**Definition 3** (Admissible MPA). *Given auxiliary variable $u$, the function $\boldsymbol{h}(\cdot)$ is said to be an admissible MPA of $\{\mathbb{P}_{\boldsymbol{x}|u=u_i}\}_{i=1}^{I}$ if and only if $\mathbb{P}_{\boldsymbol{x}|u=u_i} = \boldsymbol{h}_{\#\mathbb{P}_{\boldsymbol{x}|u=u_i}}, \forall i \in [I]$.*

Now, due to the deterministic relationship between the pair $\boldsymbol{x}$ and $\boldsymbol{y}$, we have the following fact:

**Fact 2.** *Suppose that Assumption 1 holds. Then, there exists an admissible MPA of $\{\mathbb{P}_{\boldsymbol{x}|u=u_i}\}_{i=1}^{I}$ if and only if there exists an admissible MPA of $\{\mathbb{P}_{\boldsymbol{y}|u=u_i}\}_{i=1}^{I}$.*

The above means that if we establish that there is no admissible MPA of the $\{\mathbb{P}_{\boldsymbol{x}|u=u_i}\}_{i=1}^{I}$, it suffices to conclude that there is no admissible MPA of $\{\mathbb{P}_{\boldsymbol{y}|u=u_i}\}_{i=1}^{I}\}$.

As described before, to ensure identifiability of the translation functions via solving the `DIMENSION` loss, we hope the conditional distributions $\mathbb{P}_{\boldsymbol{x}|u=u_i}$ and $\mathbb{P}_{\boldsymbol{y}|u=u_i}$ to be sufficiently different. We formalize this requirement in the following definition:

**Definition 4** (Sufficiently Diverse Condition (SDC)). *For any two disjoint sets $\mathcal{A}, \mathcal{B} \subset \mathcal{X}$, where $\mathcal{A}$ and $\mathcal{B}$ are connected, open, and non-empty, there exists a $u_{(\mathcal{A},\mathcal{B})} \in \{u_1, \ldots, u_I\}$ such that $\mathbb{P}_{\boldsymbol{x}|u=u_{(\mathcal{A},\mathcal{B})}}[\mathcal{A}] \neq \mathbb{P}_{\boldsymbol{x}|u=u_{(\mathcal{A},\mathcal{B})}}[\mathcal{B}]$. Then, the set of conditional distributions $\{\mathbb{P}_{\boldsymbol{x}|u=u_i}\}_{i=1}^{I}$ is called sufficiently diverse.*

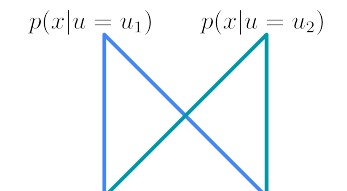

Figure 5: Conditional PDFs $p(x|u = u_1)$ and $p(x|u = u_2)$ that satisfy the SDC.

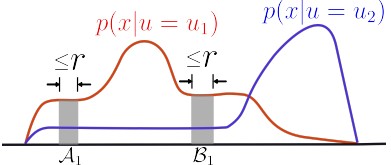

Figure 6: Illustration of relaxed SDC ($r$-SDC).

Definition 4 puts the desired "diversity" into context. It is important to note that the SDC only requires the *existence* of a certain $u_{(\mathcal{A},\mathcal{B})} \in \{u_1, \ldots, u_I\}$ for a given disjoint set pair $(\mathcal{A}, \mathcal{B})$. It does not require a unified $u$ for all pairs; i.e., $u_{(\mathcal{A},\mathcal{B})}$ needs not to be the same as $u_{(\mathcal{A}',\mathcal{B}')}$ for $(\mathcal{A}, \mathcal{B}) \neq (\mathcal{A}', \mathcal{B}')$. Fig. 5 shows a simple example where the two conditional distributions satisfy the SDC. In more general cases, this implies that if the PDFs of the conditional distributions exhibit different "shapes" over their supports, SDC is likely to hold. Using SDC, we show the following translation identifiability result:

**Theorem 1** (Identifiability). *Suppose that Assumption 1 holds. Let $\mathsf{E}_{i,j}$ denote the event that the pair $(\mathbb{P}_{\boldsymbol{x}|u=u_i}, \mathbb{P}_{\boldsymbol{x}|u=u_j})$ does not satisfy the SDC. Assume that $\Pr[\mathsf{E}_{i,j}] \leq \rho$ for any $i \neq j$, where $i, j \in [I]$. Let $(\widehat{\boldsymbol{f}}, \widehat{\boldsymbol{g}})$ be from an optimal solution of the `DIMENSION` loss (7). Then, there is no admissible MPA of $\{\mathbb{P}_{\boldsymbol{x}|u=u_i}\}_{i=1}^{I}$ of the solution, i.e., $\widehat{\boldsymbol{f}} = \boldsymbol{f}^{\star}$, a.e. and $\widehat{\boldsymbol{g}} = \boldsymbol{g}^{\star}$, a.e. with a probability of at least $1 - \rho^{\binom{I}{2}}$.*

Theorem 1 shows that if the conditional distributions are sufficiently diverse, solving (7) can correctly identify the ground-truth translation functions. Theorem 1 also spells out the importance of having more $u_i$'s (which means more auxiliary information). The increase of $I$ improves the probability of success quickly.

**Towards More Robust Identifiability.** Theorem 1 uses the fact that the SDC holds with high probability for every pair of $(\mathbb{P}_{\boldsymbol{x}|u_i}, \mathbb{P}_{\boldsymbol{x}|u_j})$ (cf. $\Pr[\mathsf{E}_{i,j}] \leq \rho$). It is also of interest to see if the method is robust to violation of the SDC. To this end, consider the following condition:

**Definition 5** (Relaxed Condition: $r$-SDC). *Let $\mathrm{dia}(\mathcal{A}) = \sup_{\boldsymbol{w},\boldsymbol{z}\in\mathcal{A}}\|\boldsymbol{w} - \boldsymbol{z}\|_2$ and $\mathcal{V}_{i,j} = \{(\mathcal{A}, \mathcal{B}) \mid \mathbb{P}_{\boldsymbol{x}|u_i}[\mathcal{A}] = \mathbb{P}_{\boldsymbol{x}|u_i}[\mathcal{B}] \;\&\; \mathbb{P}_{\boldsymbol{x}|u_j}[\mathcal{A}] = \mathbb{P}_{\boldsymbol{x}|u_j}[\mathcal{B}], \mathcal{A} \cap \mathcal{B} = \phi\}$, where $\mathcal{A}, \mathcal{B}$ are non-empty, open and connected. Denote $M_{i,j} = \max_{(\mathcal{A},\mathcal{B})\in\mathcal{V}_{i,j}} \max\{\mathrm{dia}(\mathcal{A}), \mathrm{dia}(\mathcal{B})\}$. Then, $(\mathbb{P}_{\boldsymbol{x}|u_i}, \mathbb{P}_{\boldsymbol{x}|u_j})$ satisfies the $r$-SDC if $M_{i,j} \leq r$ for $r \geq 0$.*

Note that the $r$-SDC becomes the SDC when $r = 0$. Unlike SDC in Definition 4, the relaxed SDC condition allows the violation of SDC over regions $\mathcal{V}_{i,j}$. Our next theorem shows that the translation identifiability still approximately holds, as long as the largest region in $\mathcal{V}_{i,j}$ is not substantial:

**Theorem 2** (Robust Identifiability). *Suppose that Assumption 1 holds with $\boldsymbol{g}^\star$ being L-Lipschitz continuous, and that any pair of $(\mathbb{P}_{\boldsymbol{x}|u_i}, \mathbb{P}_{\boldsymbol{x}|u_j})$ satisfies the $r$-SDC (cf. Definition 5) with probability at least $1 - \gamma$, i.e., $\Pr[M_{i,j} \geq r] \leq \gamma$ for any $i \neq j$, where $(i, j) \in [I] \times [J]$. Let $\widehat{\boldsymbol{g}}$ be from any optimal solution of the* `DIMENSION` *loss in (7). Then, we have $\|\widehat{\boldsymbol{g}}(\boldsymbol{x}) - \boldsymbol{g}^\star(\boldsymbol{x})\|_2 \leq 2rL, \quad \forall \boldsymbol{x} \in \mathcal{X}$, with a probability of at least $1 - \gamma^{\binom{I}{2}}$. The same holds for $\widehat{\boldsymbol{f}}$.*

Theorem 2 asserts that the estimation error of $\widehat{\boldsymbol{g}}$ scales linearly with the "degree" of violation of the SDC (measured by $r$). The result is encouraging: It shows that even if the SDC is violated, the performance of `DIMENSION` will not decline drastically. The Lipschitz continuity assumption in Theorem 2 is mild. Note that translation functions are often represented by neural networks in practice, and neural networks with bounded weights are Lipschitz continuous functions (Bartlett et al., 2017). Hence, the numerical successes of many neural UDT models (e.g., CycleGAN) suggest that assuming that Lipschitz continuous ground-truth translation functions exist is reasonable.

## 4 RELATED WORKS

Prior to CycleGAN (Zhu et al., 2017), the early works (Liu & Tuzel, 2016; Taigman et al., 2017; Kim et al., 2017) started using GAN-based neural structures for distribution matching in the context of I2I translation. Similar ideas appeared in UDT problems in NLP (e.g., machine translation) (Conneau et al., 2017; Lample et al., 2017). In the literature, it was noticed that distribution matching modules lack solution uniqueness, and many works proposed remedies (see, e.g, (Liu et al., 2017; Xu et al., 2022; Xie et al., 2022; Park et al., 2020)). These approaches have worked to various extents empirically, but the translation identifiability question was unanswered. The term "content" was used in the vision literature (in the context of I2I translation) to refer to domain-invariant attributes (e.g., pose and orientation (Kim et al., 2020; Amodio & Krishnaswamy, 2019; Wu et al., 2019; Yang et al., 2023)). This is a narrower interpretation of content relative to ours—as content in our case can be high-level or latent semantic meaning that is not represented by specific attributes. Our definition of content is closer to that in multimodal and self-supervised learning (Von Kügelgen et al., 2021; Lyu et al., 2022; Daunhawer et al., 2023). Before our work, auxiliary information was also considered in UDT. For example, semi-supervised UDT (see, e.g., (Wang et al., 2020; Mustafa & Mantiuk, 2020)) uses a small set of paired data samples, but our method does not use any sample-level pairing information. Attribute-guided I2I translation (see, e.g., (Li et al., 2019; Choi et al., 2018; 2020)) specifies the desired attributes in the target domain to "guide" the translation. These are different from our auxiliary variables that can be both sample attributes or high-level concepts (which is closer to the "auxiliary variables" in nonlinear independent component analysis works, e.g., (Hyvarinen et al., 2019)). Again, translation identifiability was not considered for semi-supervised or attribute-guided UDT. There has been efforts towards understanding the translation identifiability of CycleGAN. The works of Galanti et al. (2018b;a) recognized that the success of UDT may attribute to the existence of a small number of MPAs. Moriakov et al. (2020) showed that MPA exists in the solution space of CycleGAN, and used it to explain the ill-posedness of CycleGAN. Chakrabarty & Das (2022) studied the finite sample complexity of CycleGAN in terms of distribution matching and cycle consistency. Gulrajani & Hashimoto (2022) and de Bézenac et al. (2021) argued that if the target translation functions have known structures (e.g., linear or optimal transport structures), then translation identifiability can be established. However, these conditions can be restrictive. Translation identifiability without using such structural assumptions had remained unclear before our work.

## 5 NUMERICAL VALIDATION

**Constructing Challenging Translation Tasks.** We construct challenging translation tasks to validate our theorems and to illustrate the importance of translation identifiability. To this end, we make three datasets. The first two are "MNIST v.s. Rotated MNIST" (MrM) and "Edges v.s. Rotated Shoes" (ErS). In both datasets, the rotated domains consist of samples from the "MNIST" and "Shoes" with a 90 degree rotation, respectively. We intentionally make this rotation, as rotation is

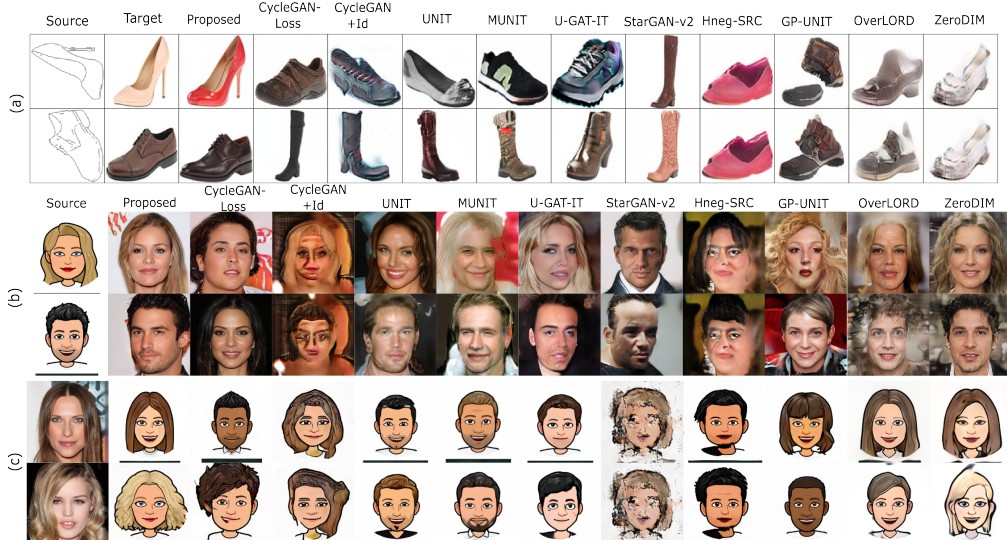

Figure 8: Qualitative results on (a) Edges to Rotated Shoes, (b) Bitmoji Faces to CelebA-HQ, and (c) CelebA-HQ to Bitmoji Faces tasks. More comprehensive illustrations are in the appendix.

a large geometric change across domains. This type of large geometric change poses a challenging translation task (Kim et al., 2020; Wu et al., 2019; Amodio & Krishnaswamy, 2019; Yang et al., 2023). In addition, we construct a task "CelebA-HQ (Karras et al., 2017) v.s. Bitmoji (Mozafari, 2020)" (CB). In this task, profile photos of celebrities are translated to cartoonized bitmoji figures, and vice versa. We intentionally choose these two domains to make the translation challenging: The profile photos have rich details and are diverse in terms of face orientation, expression, hair style, etc., but the Bitmoji pictures have a relatively small set of choices of these attributes (e.g., they are always front-facing). More details of the datasets are in Sec. F.4 in the supplementary material.

**Baselines.** The baselines include some representative UDT methods and some recent developments, i.e., `GP-UNIT` (Yang et al., 2023), `Hneg-SRC` (Jung et al., 2022), `OverLORD` (Gabbay & Hoshen, 2021), `ZeroDIM` (Gabbay et al., 2021), `StarGAN-v2` (Choi et al., 2020), `U-GAT-IT` (Kim et al., 2020), `MUNIT` (Huang et al., 2018), `UNIT` (Liu et al., 2017), and `CycleGAN` (Zhu et al., 2017). In particular, two versions of CycleGAN are used. "`CycleGAN Loss`" refers to the plan-vanilla CycleGAN objective in (3) and `CycleGAN+Id` refers to the "identity-regularized" version in (Zhu et al., 2017). `ZeroDIM` uses the same auxiliary information as that used by the proposed method.

**MNIST to Rotated MNIST.** Fig. 7 shows the results. In this case, we use $u \in \{1, \dots, 10\}$, i.e., the labels of the identity of digits, as the alphabet of the auxiliary variable. Note that knowing such labels does not mean that the cross-domain pairs $(\boldsymbol{x}, \boldsymbol{y})$ are known. Alternatively, one can also use digit shapes as

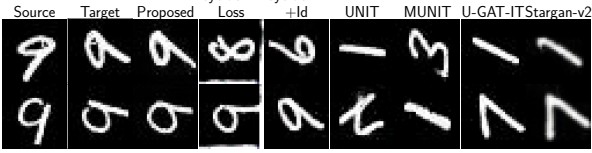

Figure 7: Translation from MNIST to rotated MNIST.

the alphabets (see Sec. F.6). One can see that `DIMENSION` learns to translate the digits to their corresponding rotated versions. But the baselines sometimes misalign the samples. The results are consistent with our analysis (see Sec. F.6 for more results).

**Edges to Rotated Shoes.** From Fig. 8 (a), one can see that the baselines all misalign the edges with wrong shoes. Instead, the proposed `DIMENSION`, using the shoe types (shoes, boots, sandals, and slippers) as the alphabet of $u$, does not encounter this issue. More experiments including the reverse translation (i.e., shoes to edges) are in Sec. F.6 in the supplementary material.

**CelebA-HQ and Bitmoji.** Figs. 8 (b)-(c) show the results. The proposed method uses $u \in \{$'`male`','`female`','`black hair`','`non-black hair`'$\}$. To obtain the auxilliary information for each sample, we use CLIP to automatically annotate the images. A remark is that translating from the Bitmoji domain to the CelebA-HQ domain [see. Fig. 8 (b)] is particularly hard. This is because the learned translation function needs to "fill in" a lot of details to make the generated profiles

Table 1: `LPIPS` scores for the ErS and MrM tasks and `FID` scores for all tasks. E: Edges, rS: rotated Shoes, M: MNIST, rM: rotated MNIST, C: CelebA-HQ, B: Bitmoji faces.

| Method | LPIPS ($\downarrow$) | | | | FID ($\downarrow$) | | | | | |
|---|---|---|---|---|---|---|---|---|---|---|
| | E $\to$ rS | rS $\to$ E | M $\to$ rM | rM $\to$ M | E | rS | M | rM | C | B |
| Proposed | **0.29 ± 0.06** | **0.35 ± 0.10** | **0.11 ± 0.08** | **0.09 ± 0.04** | 21.47 | 40.14 | 13.95 | 16.07 | **32.03** | **20.50** |
| CycleGAN-Loss | 0.43 ± 0.06 | 0.50 ± 0.07 | 0.34 ± 0.07 | 0.33 ± 0.09 | 35.83 | 55.42 | 16.09 | 16.11 | 36.71 | 28.02 |
| CycleGAN | 0.65 ± 0.03 | 0.54 ± 0.07 | 0.27 ± 0.09 | 0.28 ± 0.09 | 259.31 | 130.84 | 46.05 | 34.01 | 196.52 | 85.05 |
| U-GAT-IT | 0.56 ± 0.05 | 0.48 ± 0.07 | 0.25 ± 0.09 | 0.25 ± 0.09 | 288.03 | 58.20 | **11.78** | **11.67** | 50.28 | 39.09 |
| UNIT | 0.49 ± 0.03 | 0.58 ± 0.03 | 0.25 ± 0.06 | 0.25 ± 0.08 | 33.95 | 96.28 | 20.44 | 19.15 | 53.63 | 33.56 |
| MUNIT | 0.50 ± 0.03 | 0.58 ± 0.04. | 0.28 ± 0.09 | 0.28 ± 0.09 | 43.83 | 86.68 | 14.89 | 15.96 | 62.49 | 27.59 |
| StarGAN-v2 | 0.39 ± 0.05 | 0.52 ± 0.11 | 0.28 ± 0.09 | 0.29 ± 0.10 | 75.46 | 138.34 | 30.07 | 32.20 | 35.44 | 282.98 |
| Hneg-SRC | 0.45 ± 0.06 | 0.50 ± 0.07 | – | – | 210.27 | 198.77 | – | – | 129.34 | 66.36 |
| GP-UNIT | 0.49 ± 0.08 | 0.44 ± 0.05 | – | – | 231.31 | 96.32 | – | – | 32.40 | 30.30 |
| OverLORD | 0.43 ± 0.06 | 0.42 ± 0.05 | – | – | 101.14 | 124.02 | – | – | 76.10 | 31.08 |
| ZeroDIM | 0.38 ± 0.06 | 0.41 ± 0.07 | – | – | 85.56 | 187.45 | – | – | 88.36 | 36.21 |

"–" means that method is not applicable to the dataset due to small resolution.

photorealistic. Our method clearly outperforms the baselines in both directions of translation; see more in Sec. F.6 in the supplemenary material.

**Metrics and Quantative Evaluation.** We employ two widely adopted metrics in UDT. The first is the *learned perceptual image patch similarity* (`LPIPS`) (Zhang et al., 2018), which leverages the known ground-truth correspondence between $(x, y)$. `LPIPS` measures the "perceptual distance" between the translated images and the ground-truth target images. In addition, we also use the *Fréchet inception distance* (`FID`) score (Heusel et al., 2017) in all tasks. `FID` measures the visual quality of the learned translation using a distribution divergence between the translated images and the target domain. In short, `LPIPS` and `FID` correspond to the content alignment performance and the target domain-attaining ability, respectively; see details of the metrics Sec. F.4.

Table 1 shows the `LPIPS` scores over the first two datasets where the ground-truth pairs are known. One can see that `DIMENSION` significantly outperforms the baselines—which is a result of good content alignment. The `FID` scores in the same table show that our method produces translated images that have similar characteristics of the target domains. The `FID` scores output by our method are either the lowest or the second lowest.

**Detailed Settings and More Experiments.** See Sec. E-H for settings and more results.

## 6 CONCLUSION

In this work, we revisited the UDT and took a deep look at a core theoretical challenge, namely, the translation identifiability issue. Existing UDT approaches (such as CycleGAN) often lack translation identifiability and may produce content-misaligned translations. This issue largely attributes to the presence of MPA in the solution space of their distribution matching modules. Our approach leverages the existence of domain-invariant auxiliary variables to establish translation identifiability, using a novel diversified distribution matching criterion. To our best knowledge, the identifiability result stands as the first of its kind, without using restrictive conditions on the structure of the desired translation functions. We also analyzed the robustness of proposed method when the key sufficient condition for identifiability is violated. Our identifiability theory leads to an easy-to-implement UDT system. Synthetic and real-data experiments corroborated with our theoretical findings.

**Limitations.** Our work considers a model where the ground-truth translation functions are deterministic and bijective. This setting has been (implicitly or explicitly) adopted by a large number of existing works, with the most notable representative being CycleGAN. However, there can be multiple "correct" translation functions in UDT, as the same "content" can be combined with various "styles". Such cases may be modeled using probabilistic translation mechanisms (Huang et al., 2018; Choi et al., 2020; Yang et al., 2023), yet the current analytical framework needs a significant revision to accommodate the probabilistic setting. In addition, our method makes use of auxiliary variables that may be nontrivial to acquire in certain cases. We have shown that open-sourced foundation models such as CLIP can help acquire such auxiliary variables and that the method is robust to noisy/wrong auxiliary variables (see Sec. H). However, it is still of great interest to develop provable UDT translation schemes without using auxiliary variables.

**Acknowledgement.** This work is supported in part by the Army Research Office (ARO) under Project ARO W911NF-21-1-0227, and in part by the National Science Foundation (NSF) CAREER Award ECCS-2144889.

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

**Supplementary Material of "Towards Identifiable Unsupervised Domain Translation: A Diversified Distribution Matching Approach"**

## A    PRELIMINARIES

### A.1    NOTATION

- $x$, $\boldsymbol{x}$, $\mathcal{X}$ denote a scalar, vector, and a set, respectively.
- $p(\boldsymbol{x})$ and $p(\boldsymbol{x}|u)$ denote the marginal *probability density function* (PDF) of $\boldsymbol{x}$ and conditional PDF of $\boldsymbol{x}$ conditioned on $u$, respectively.
- $\|\boldsymbol{x}\|_2$ denotes the $\ell_2$-norm of $\boldsymbol{x}$.
- $\mathrm{dia}(\mathcal{A}) = \sup_{\boldsymbol{a},\boldsymbol{b}\in\mathcal{A}} \|\boldsymbol{a}-\boldsymbol{b}\|_2$.
- $\mathbb{I}: \mathcal{X} \to \mathcal{X}$ denotes the identity function such that $\mathbb{I}(\boldsymbol{x}) = \boldsymbol{x}, \forall \boldsymbol{x} \in \mathcal{X}$.
- $\mathcal{A}^c$, $\mathrm{cl}(\mathcal{A})$, $\mathrm{bd}(\mathcal{A})$ and $\mathrm{int}(\mathcal{A})$ denote the complement, closure, boundary, and the interior of set $\mathcal{A}$.
- A set $\mathcal{A}$ is said to have strictly positive measure under $p(\boldsymbol{x})$ if and only if $\mathbb{P}_{\boldsymbol{x}}[\mathcal{A}] > 0$.
- For a (random) vector $\boldsymbol{x}$, $x(i)$ and $[\boldsymbol{x}]_i$ denote the $i$th element of $\boldsymbol{x}$, and $\boldsymbol{x}(i:j)$ denotes $[x(i), x(i+1), \ldots, x(j)]$.
- Distance between two sets is defined as

$$\mathrm{dist}(\mathcal{A},\mathcal{B}) = \inf_{\boldsymbol{a}\in\mathcal{A},\boldsymbol{b}\in\mathcal{B}} \|\boldsymbol{a}-\boldsymbol{b}\|_2.$$

- Distance between a set and a point is defined as

$$\mathrm{dist}(\boldsymbol{a},\mathcal{B}) = \inf_{\boldsymbol{b}\in\mathcal{B}} \|\boldsymbol{a}-\boldsymbol{b}\|_2.$$

- $\mathcal{N}_\epsilon(\boldsymbol{z})$ denotes the $\epsilon$-neighborhood of $\boldsymbol{z} \in \mathbb{R}^N$ defined as

$$\mathcal{N}_\epsilon(\boldsymbol{z}) = \{\widehat{\boldsymbol{z}} \in \mathbb{R}^N | \|\boldsymbol{z}-\widehat{\boldsymbol{z}}\|_2 < \epsilon\}.$$

- $\mathrm{conn}(\mathcal{A})$ denotes the set of connected components of $\mathcal{A}$ (see definition of connected components in Appendix A.2).
- For any function $\boldsymbol{m}: \mathcal{W} \to \mathcal{Z}$, and set $\mathcal{A} \subseteq \mathcal{W}$, $\boldsymbol{m}(\mathcal{A}) = \{\boldsymbol{m}(\boldsymbol{w}) \in \mathcal{Z} \mid \boldsymbol{w} \in \mathcal{A}\}$

### A.2    DEFINITIONS

We will employ standard notions from real analysis. We refer the readers to (Carothers, 2000; Rudin, 1976) for precise definitions and more details. Here we provide working definition with illustration.

**Connected set.** A set $\mathcal{C}$ is connected (in $\mathcal{X}$), if and only if there does not exist any disjoint non-empty open sets $\mathcal{A}, \mathcal{B} \subset \mathcal{X}$ such that $\mathcal{A} \cap \mathcal{C} \neq \phi$, $\mathcal{B} \cap \mathcal{C} \neq \phi$, and $\mathcal{C} \subset \mathcal{A} \cup \mathcal{B}$ (see Fig. 9).

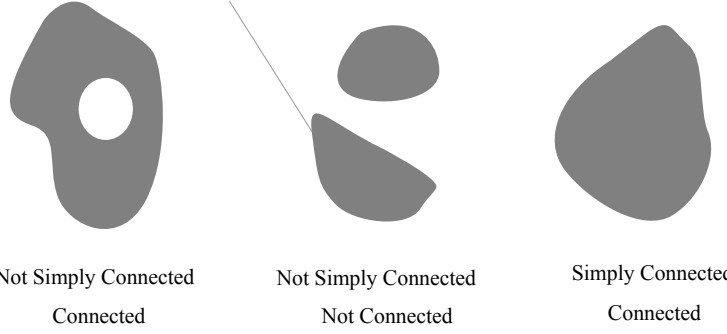

Figure 9: Illustration of connected and simply connected sets

**Simply connected set.** A simply connected set is a connected set such that any simple closed curve can be shrunk to a point continuously in the set (see Fig. 9).

**Connected components.** Given a set $\mathcal{A}$, the maximal connected subsets of $\mathcal{A}$, such that the subsets are not themselves contained in any other connected subsets of $\mathcal{A}$, are called connected components of $\mathcal{A}$. Specifically, a connected set $\mathcal{C} \subseteq \mathcal{A}$ is a connected component of $\mathcal{A}$ if there does not exist any other connected set $\mathcal{D} \subseteq \mathcal{A}$, such that $\mathcal{C} \subset \mathcal{D}$. In Fig. 10, $\mathcal{A}$ denotes the entire shaded regions, and has three connected components $\mathcal{C}_1$, $\mathcal{C}_2$, and $\mathcal{C}_3$. Note that any set can be uniquely written as a disjoint union of its connected components. In Fig. 10, $\mathcal{C}_1 \cup \mathcal{C}_2 \cup \mathcal{C}_3$ is a unique disjoint union representing $\mathcal{A}$.

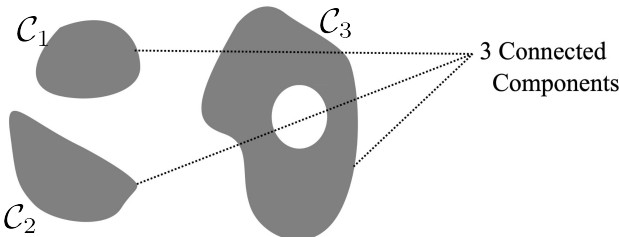

Figure 10: A set $\mathcal{A} = \mathcal{C}_1 \cup \mathcal{C}_2 \cup \mathcal{C}_3$ with 3 connected components: $\mathcal{C}_1, \mathcal{C}_2$, and $\mathcal{C}_3$.

**Continuous function.** A function $\boldsymbol{m} : \mathcal{W} \to \mathcal{Z}$, with $\mathcal{W} \subseteq \mathbb{R}^W, \mathcal{Z} \subseteq \mathbb{R}^Z$ is said to be continuous if for any $\boldsymbol{w} \in \mathcal{W}$ and $\epsilon > 0$, there exists a $\delta > 0$ such that

$$\boldsymbol{m}\left(\left(\mathcal{N}_\delta(\boldsymbol{w}) \cap \mathcal{W}\right)\right) \subset \mathcal{N}_\epsilon(\boldsymbol{m}(\boldsymbol{w})) \cap \mathcal{Z}.$$

**Continuous and invertible Functions.** If a function $\boldsymbol{m} : \mathcal{W} \to \mathcal{Z}$ is continuous and invertible, then its inverse $\boldsymbol{m}^{-1} : \mathcal{Z} \to \mathcal{W}$ is also continuous. Some useful properties of continuous and invertible function $\boldsymbol{m}$ are as follows:

- If $\mathcal{A} \subseteq \mathcal{W}$ is closed, then $\boldsymbol{m}(\mathcal{A})$ is also closed.
- If $\mathcal{A} \subseteq \mathcal{W}$ is open, then $\boldsymbol{m}(\mathcal{A})$ is also open.
- If $\mathcal{A} \subseteq \mathcal{W}$ is connected, then $\boldsymbol{m}(\mathcal{A})$ is also connected.

# B  PROOF OF LEMMAS AND FACTS

Note that for the ease of reading, the lemmas, facts, and theorems from the main paper are re-stated and highlighted using shaded boxes.

> **Proposition 1.** *Suppose that $\mathbb{P}_{\boldsymbol{x}}$ admits a continuous PDF, $p(\boldsymbol{x})$ and $p(\boldsymbol{x}) > 0, \forall \boldsymbol{x} \in \mathcal{X}$. Assume that $\mathcal{X}$ is simply connected. Then, there exists a continuous non-trivial (non-identity) $\boldsymbol{h}(\cdot)$ such that $\boldsymbol{h}_{\#\mathbb{P}_{\boldsymbol{x}}} = \mathbb{P}_{\boldsymbol{x}}$.*

*Proof.* We want to show that there exists a continuous $\boldsymbol{h} : \mathcal{X} \to \mathcal{X}$ such that

$$\boldsymbol{h}_{\#\mathbb{P}_{\boldsymbol{x}}} = \mathbb{P}_{\boldsymbol{x}}.$$

To this end, we will construct such MPA by reducing the problem of finding an MPA of $p(\boldsymbol{x})$ to finding an MPA of the uniform distribution. Note that one can always construct a continuous invertible function $\boldsymbol{d} : \mathcal{X} \to (0, 1)^{D_x}$, such that the function maps any continuous distribution with a simply connected support to the uniform distribution. This mapping can be found via the so-called *Darmois construction* (Darmois, 1951; Hyvärinen & Pajunen, 1999). Specifically, under the Darmois construction, the $i$th output of $\boldsymbol{d}(\overline{\boldsymbol{x}}), \forall \overline{\boldsymbol{x}} \in \mathcal{X}$ is given by

$$[\boldsymbol{d}(\overline{\boldsymbol{x}})]_i := F\left(\overline{x}(i) \mid \boldsymbol{x}(1 : i - 1) = \overline{\boldsymbol{x}}(1 : i - 1)\right), \quad i = 1, \ldots, N,$$

where $F\left(\overline{x}(i) \mid \cdot\right)$ denotes the conditional CDF of $x(i)$, i.e,

$$F\left(\overline{x}(i) \mid \boldsymbol{x}(1 : i - 1) = \overline{\boldsymbol{x}}(1 : i - 1)\right) = \mathbb{P}_{x(i)|\boldsymbol{x}(1:i-1)=\overline{\boldsymbol{x}}(1:i-1)}\left[\{x(i) : x(i) \leq \overline{x}(i)\}\right];$$

see more detailed introduction to the Darmois construction in (Hyvärinen & Pajunen, 1999).

With the constructed $d$, one can form a continuous mapping $h : \mathcal{X} \to \mathcal{X}$ as follows

$$h = d^{-1} \circ h_U \circ d,$$

where $h_U : (0,1)^{D_x} \to (0,1)^{D_x}$ is a continuous MPA on the uniform distribution over $(0,1)^{D_x}$. Since $d$ is continuous for a continuous distribution, $h$ is continuous because it is the composition of continuous functions.

Now, it remains to show that $h_U$ exists. A simple example of $h_U$ is reflection around the mean of $d(x)$, i.e.,

$$h_U(z) = -z + 2\mu,$$

where $\mu = [1/2, \ldots, 1/2]^\top \in \mathbb{R}^{D_x}$. This concludes the proof. $\qquad\square$

**Lemma 1.** *Assume that an optimal solution of (7) is $(\widehat{f}, \widehat{g}, \{\widehat{d}_x^{(i)}, \widehat{d}_y^{(i)}\})$. Then, under Assumption 1, we have $\mathbb{P}_{x|u=u_i} = \widehat{f}_{\#}\mathbb{P}_{y|u=u_i}$, $\mathbb{P}_{y|u=u_i} = \widehat{g}_{\#}\mathbb{P}_{x|u=u_i}$, $\forall i \in [I]$, and $\widehat{f} = \widehat{g}^{-1}$, a.e.*

*Proof.* Fact 1 is a direct consequence of (Goodfellow et al., 2014, Theorem 1).

First of all, recall the objective in (7):

$$\min_{f,g} \max_{\{d_x^{(i)}, d_y^{(i)}\}} \sum_{i=1}^{I} \Big( \mathcal{L}_{\mathrm{DSGAN}}(g, d_y^{(i)}, x, y) + \mathcal{L}_{\mathrm{DSGAN}}(f, d_x^{(i)}, x, y) \Big) + \lambda \mathcal{L}_{\mathrm{cyc}}(g, f). \quad (8)$$

The global minimum of $\mathcal{L}_{\mathrm{DSGAN}}(g, d_y^{(i)}, x, y)$ is achieved when (Goodfellow et al., 2014, Theorem 1)

$$g_{\#}\mathbb{P}_{x|u=u_i} = \mathbb{P}_{y|u=u_i}.$$

Similarly, the global minimum of $\mathcal{L}_{\mathrm{DSGAN}}(f, d_x^{(i)}, x, y)$ is achieved when

$$f_{\#}\mathbb{P}_{y|u=u_i} = \mathbb{P}_{x|u=u_i}.$$

Finally, the global minimum of $\mathcal{L}_{\mathrm{cyc}}(g, f)$, which is zero, is achieved when

$$g = f^{-1}, a.e.$$

We know that $g^\star$ and $f^\star$ can achieve global minimums of all loss terms simultaneously. Hence the solution of (7), $\widehat{f}$ and $\widehat{g}$, should satisfy

$$\mathbb{P}_{x|u=u_i} = \widehat{f}_{\#}\mathbb{P}_{y|u=u_i}, \ \mathbb{P}_{y|u=u_i} = \widehat{g}_{\#}\mathbb{P}_{x|u=u_i}, \ \forall i \in [I], \text{ and } \widehat{f} = \widehat{g}^{-1}, a.e.$$

$\qquad\square$

**Fact 2.** *Suppose that Assumption 1 holds. Then, there exists an admissible MPA of $\{\mathbb{P}_{x|u=u_i}\}_{i=1}^{I}$ if and only if there exists an admissible MPA of $\{\mathbb{P}_{y|u=u_i}\}_{i=1}^{I}$.*

*Proof.* Let $h$ be an admissible MPA of $\{\mathbb{P}_{x|u=u_i}\}_{i=1}^{I}$. Then

$$\begin{aligned}
h_{\#}\mathbb{P}_{x|u=u_i} &= \mathbb{P}_{x|u=u_i}, \forall i \in [I]. \\
\iff g^\star \circ h_{\#}\mathbb{P}_{x|u=u_i} &= g^\star_{\#}\mathbb{P}_{x|u=u_i}, \forall i \in [I]. \\
\iff g^\star \circ h \circ f^\star_{\#}\mathbb{P}_{y|u=u_i} &= \mathbb{P}_{y|u=u_i}, \forall i \in [I].
\end{aligned}$$

This implies that $g^\star \circ h \circ f^\star$ is an admissible MPA of $\{\mathbb{P}_{y|u=u_i}\}_{i=1}^{I}$ if and only if $h$ is an admissible MPA of $\{\mathbb{P}_{x|u=u_i}\}_{i=1}^{I}$.

Hence, there exists an admissible MPA of $\{\mathbb{P}_{x|u=u_i}\}_{i=1}^{I}$ if and only if there exists an admissible MPA of $\{\mathbb{P}_{y|u=u_i}\}_{i=1}^{I}$. $\qquad\square$

## C  PROOF OF THEOREMS

### C.1  PROOF OF THEOREM 1

**Theorem 1.** *Suppose that Assumption 1 holds. Let $\mathsf{E}_{i,j}$ denote the event that the set $\{\mathbb{P}_{\boldsymbol{x}|u=u_i}, \mathbb{P}_{\boldsymbol{x}|u=u_j}\}$ does not satisfy the SDC. Assume that $\Pr[\mathsf{E}_{i,j}] \leq \rho$ for any $i \neq j$, where $i, j \in [I]$. Let $(\widehat{\boldsymbol{f}}, \widehat{\boldsymbol{g}})$ be from an optimal solution of the* DIMENSION *loss* (7). *Then, there is no admissible MPA of $\{\mathbb{P}_{\boldsymbol{x}|u=u_i}\}_{i=1}^{I}$ of the solution, i.e., $\widehat{\boldsymbol{f}} = \boldsymbol{f}^{\star}$, a.e., $\widehat{\boldsymbol{g}} = \boldsymbol{g}^{\star}$, a.e., with a probability of at least $1 - \rho^{\binom{I}{2}}$.*

Theorem 1 is a direct consequence of following lemma:

**Lemma A.1.** *Suppose that Assumption 1 holds. Assume that $\{\mathbb{P}_{\boldsymbol{x}|u=u_i}\}_{i=1}^{I}$ are sufficiently diverse. Then, $\widehat{\boldsymbol{g}} = \boldsymbol{g}^{\star}$ and $\widehat{\boldsymbol{f}} = \boldsymbol{f}^{\star}$, a.e.*

*Proof of Lemma A.1.* First, we show that no non-trivial continuous admissible MPA exists for $\{\mathbb{P}_{\boldsymbol{x}|u=u_i}\}_{i=1}^{I}$, i.e., if a continuous $\boldsymbol{h}$ satisfies

$$\boldsymbol{h}_{\#\mathbb{P}_{\boldsymbol{x}|u=u_i}} = \mathbb{P}_{\boldsymbol{x}|u=u_i} \forall i \in [I], \tag{9}$$

then $\boldsymbol{h} = \mathbb{I}$, a.e.

Eq. (9), by the definition of push-forward measure, implies that

$$\implies \mathbb{P}_{\boldsymbol{x}|u_i}[\boldsymbol{h}(\mathcal{A})] = \mathbb{P}_{\boldsymbol{x}|u_i}[\mathcal{A}], \forall i \in [I]. \tag{10}$$

For the sake of contradiction assume that $\boldsymbol{h}$ satisfies (9), however, $\boldsymbol{h} \neq \mathbb{I}$ on a set of strictly positive measure. This means that there exists a $\overline{\boldsymbol{x}} \in \mathcal{X}$ such that

$$\boldsymbol{h}(\overline{\boldsymbol{x}}) \neq \overline{\boldsymbol{x}}.$$

Now, let us define an open set around $\overline{\boldsymbol{x}}$ denoted by $\mathcal{D}$ such that

$$\mathcal{D} = \mathcal{N}_d(\overline{\boldsymbol{x}}) \cap \mathcal{X}.$$

Because of the continuity and invertibility of $\boldsymbol{h}$, $\boldsymbol{h}(\mathcal{D}) \subseteq \mathcal{X}$ is also an open set and

$$\boldsymbol{h}(\overline{\boldsymbol{x}}) \in \boldsymbol{h}(\mathcal{D}).$$

Now, one can select $d$ to be small enough (because of the continuity of $\boldsymbol{h}$) such that $\mathcal{D} \cap \boldsymbol{h}(\mathcal{D}) = \phi$ and $\mathcal{D}$ is a connected set. $\mathcal{D}$ being a connected set implies that $\boldsymbol{h}(\mathcal{D})$ is also connected.

The above is a contradiction to Assumption 4 since $\mathcal{D}$ and $\boldsymbol{h}(\mathcal{D})$ are two disjoint, open and connected sets which satisfy

$$\mathbb{P}_{\boldsymbol{x}|u_i}[\boldsymbol{h}(\mathcal{D})] = \mathbb{P}_{\boldsymbol{x}|u_i}[\mathcal{D}], \forall i \in [I].$$

Hence, any $\boldsymbol{h}$ that satisfy $\boldsymbol{h}_{\#\mathbb{P}_{\boldsymbol{x}|u=u_i}} = \mathbb{P}_{\boldsymbol{x}|u=u_i}$ is such that $\boldsymbol{h} = \mathbb{I}$, a.e.

Finally, We want to show that $\widehat{\boldsymbol{g}} = \boldsymbol{g}^{\star}$, a.e. Lemma 1 implies that

$$\widehat{\boldsymbol{g}}_{\#\mathbb{P}_{\boldsymbol{x}|u=u_i}} = \mathbb{P}_{\boldsymbol{y}|u=u_i}, \forall i \in [I]$$

$$\implies \widehat{\boldsymbol{g}}_{\#\mathbb{P}_{\boldsymbol{x}|u=u_i}} = \boldsymbol{g}^{\star}_{\#\mathbb{P}_{\boldsymbol{x}|u=u_i}}, \forall i \in [I]$$

$$\overset{(a)}{\implies} \boldsymbol{g}^{\star-1} \circ \widehat{\boldsymbol{g}}_{\#\mathbb{P}_{\boldsymbol{x}|u=u_i}} = \mathbb{P}_{\boldsymbol{x}|u=u_i}, \forall i \in [I] \tag{11}$$

where (a) is obtained by applying $\boldsymbol{g}^{\star-1}$ on both sides, which is allowed because applying the same function preserves the equivalence of the distributions.

Eq. (11) implies that $\boldsymbol{g}^{\star} \circ \widehat{\boldsymbol{g}}$ is a continuous admissible MPA of $\{\mathbb{P}_{\boldsymbol{x}|u=u_i}\}_{i=1}^{I}$, which means that the following has to hold:

$$\boldsymbol{g}^{\star-1} \circ \widehat{\boldsymbol{g}} = \mathbb{I}, a.e.$$

Therefore, we always have $\widehat{\boldsymbol{g}} = \boldsymbol{g}^{\star}$, a.e. By role symmetry of $\boldsymbol{f}^{\star}$ and $\boldsymbol{g}^{\star}$ (also see Fact 2), we also have $\widehat{\boldsymbol{f}} = \boldsymbol{f}^{\star}$, a.e. $\qquad\square$

*Proof of Theorem 1.* Using the assumption that $\Pr[\mathsf{E}_{i,j}] \leq \rho$, the probability that $\{\mathbb{P}_{\boldsymbol{x}|u=u_i}\}_{i=1}^I$ are not sufficiently diverse can be bounded as follows:

$$\Pr[\{\mathbb{P}_{\boldsymbol{x}|u=u_i}\}_{i=1}^I \text{ are not sufficiently diverse}]$$

$$\overset{(a)}{\leq} \Pr\left[\bigcap_{i,j\in[I],i<j} \mathsf{E}_{i,j}\right]$$

$$\overset{(b)}{=} \bigcap_{i,j\in[I],i<j} \Pr[\mathsf{E}_{i,j}]$$

$$\leq \rho^{\binom{I}{2}},$$

where the $(a)$ holds since $\{\mathbb{P}_{\boldsymbol{x}|u=u_i}\}_{i=1}^I$ not being sufficiently diverse implies the existence of open connected sets $\mathcal{A}$ and $\mathcal{B}$ such that

$$\mathbb{P}_{\boldsymbol{x}|u_i}[\mathcal{A}] = \mathbb{P}_{\boldsymbol{x}|u_i}[\mathcal{B}], \forall i \in [I]$$
$$\implies \mathbb{P}_{\boldsymbol{x}|u_i}[\mathcal{A}] = \mathbb{P}_{\boldsymbol{x}|u_i}[\mathcal{B}], \forall i \in \{i,j\} \subset [I].$$

Finally, $(b)$ is due to the independence of the events $\mathsf{E}_{i,j}$ and $\mathsf{E}_{i,j'}$ for $j \neq j'$.

Hence, $\{\mathbb{P}_{\boldsymbol{x}|u=u_i}\}_{i=1}^I$ are sufficiently diverse with probability at least $1 - \rho^{\binom{I}{2}}$, which implies that $\widehat{\boldsymbol{f}} = \boldsymbol{f}^\star$ and $\widehat{\boldsymbol{g}} = \boldsymbol{g}^\star$ with probability at least $1 - \rho^{\binom{I}{2}}$. $\qquad\square$

## C.2 PROOF OF THEOREM 2

**Theorem 2** (Robust Identifiability). *Suppose that Assumption 1 holds with $\boldsymbol{g}^\star$ being L-Lipschitz continuous, and that any pair of $(\mathbb{P}_{\boldsymbol{x}|u_i}, \mathbb{P}_{\boldsymbol{x}|u_j})$ satisfies the $r$-SDC (cf. Definition 5) with probability at least $1 - \gamma$, i.e., $\Pr[M_{i,j} \geq r] \leq \gamma$ for any $i \neq j$, where $(i,j) \in [I] \times [J]$. Let $\widehat{\boldsymbol{g}}$ be from any optimal solution of the* `DIMENSION` *loss in (7). Then, we have $\|\widehat{\boldsymbol{g}}(\boldsymbol{x}) - \boldsymbol{g}^\star(\boldsymbol{x})\|_2 \leq 2rL, \quad \forall \boldsymbol{x} \in \mathcal{X}$, with a probability of at least $1 - \gamma^{\binom{I}{2}}$. The same holds for $\widehat{\boldsymbol{f}}$.*

First, consider the following lemma.

**Lemma A.2.** *Given any continuous admissible MPA $\boldsymbol{h}$ of $\{\mathbb{P}_{\boldsymbol{x}|u=u_i}\}_{i=1}^I$, let $\mathcal{E}_{\boldsymbol{h}}$ be a set defined as*

$$\mathcal{E}_{\boldsymbol{h}} = \{\boldsymbol{x} \mid \boldsymbol{h}(\boldsymbol{x}) \neq \boldsymbol{x}, \, \forall \boldsymbol{x} \in \mathcal{X}\}.$$

*Then, any connected component $\mathcal{C} \subseteq \mathrm{cl}(\mathcal{E}_{\boldsymbol{h}})$ satisfies*

$$\boldsymbol{x} \in \mathcal{C} \implies \boldsymbol{h}(\boldsymbol{x}) \in \mathcal{C}.$$

Lemma A.2 states an interesting property of a subset of $\mathcal{X}$ (namely, $\mathcal{E}_{\boldsymbol{h}}$) that is "modified" by the continuous MPA $\boldsymbol{h}$. Here, "modification" means that any point in the subset will land on a different point after the $\boldsymbol{h}$-transformation. The lemma shows that the source point from $\mathcal{E}_{\boldsymbol{h}}$ and its $\boldsymbol{h}$-transformation both reside in the same connected component, namely, $\mathcal{C}$. This will be useful in proving Theorem 2.

**Proof Idea:** The main idea behind the proof of Lemma A.2 is to first note that any point outside of $\mathrm{cl}(\mathcal{E}_{\boldsymbol{h}})$ is stationary under the transformation $\boldsymbol{h}$ (i.e., $\boldsymbol{h}(\boldsymbol{x}) = \boldsymbol{x}$). Next, if there was a point $\overline{\boldsymbol{x}}$ from a connected component $\mathcal{C}_1 \in \mathrm{conn}(\mathrm{cl}(\mathcal{E}_{\boldsymbol{h}}))$ was such that $\boldsymbol{h}(\overline{\boldsymbol{x}})$ was not in $\mathcal{C}_1$, then it should be either in $\mathcal{X} \backslash \mathrm{cl}(\mathcal{E}_{\boldsymbol{h}})$ or in $\mathrm{cl}(\mathcal{E}_{\boldsymbol{h}}) \backslash \mathcal{C}_1$. However, since $\boldsymbol{h}$ is invertible, $\boldsymbol{h}$ cannot map a point from $\mathcal{C}_1$ to a $\mathcal{X} \backslash \mathrm{cl}(\mathcal{E}_{\boldsymbol{h}})$. Therefore $\boldsymbol{h}(\overline{\boldsymbol{x}})$ should lie in $\mathcal{E}_{\boldsymbol{h}} \backslash \mathcal{C}_1$. But this will make the function $\boldsymbol{h}$ discontinuous. Hence, $\boldsymbol{h}(\overline{\boldsymbol{x}})$ should be in $\mathcal{C}_1$.

*Proof of Lemma A.2.* Let $\mathrm{conn}(\mathrm{cl}(\mathcal{E}_{\boldsymbol{h}}))$ denote the set of connected components of $\mathrm{cl}(\mathcal{E}_{\boldsymbol{h}})$. Suppose that there exists $\overline{\boldsymbol{x}} \in \mathcal{C}_1$ and $\mathcal{C}_1 \in \mathrm{conn}(\mathrm{cl}(\mathcal{E}_{\boldsymbol{h}}))$ such that $\boldsymbol{h}(\overline{\boldsymbol{x}}) \notin \mathcal{C}_1$. First,

$$\boldsymbol{h}(\widetilde{\boldsymbol{x}}) = \widetilde{\boldsymbol{x}}, \, \forall \widetilde{\boldsymbol{x}} \in \mathcal{X} \backslash \mathrm{cl}(\mathcal{E}_h) \overset{(a)}{\implies} \boldsymbol{h}(\overline{\boldsymbol{x}}) \neq \widetilde{\boldsymbol{x}}, \, \forall \widetilde{\boldsymbol{x}} \in \mathcal{X} \backslash \mathrm{cl}(\mathcal{E}_h)$$
$$\implies \boldsymbol{h}(\overline{\boldsymbol{x}}) \in \mathcal{C}_2, \text{ for some } \mathcal{C}_2 \in \mathrm{conn}(\mathrm{cl}(\mathcal{E}_{\boldsymbol{h}})) \backslash \mathcal{C}_1,$$

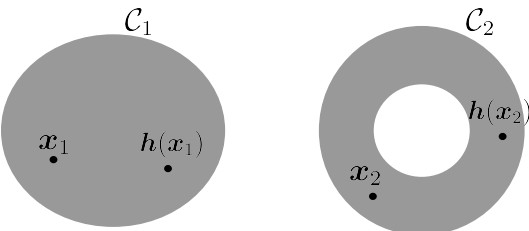

Figure 11: Illustration of Lemma A.2. $\mathcal{C}_1, \mathcal{C}_2$ are the two connected components of $\mathrm{cl}(\mathcal{E}_{\boldsymbol{h}})$. In this case, $\mathrm{cl}(\mathcal{E}_{\boldsymbol{h}}) = \mathcal{C}_1 \cup \mathcal{C}_2$. Points $\boldsymbol{x}_1$ and $\boldsymbol{x}_2$ inside $\mathcal{C}_1$ and $\mathcal{C}_2$ stay inside the same connected component after transformation by $\boldsymbol{h}$.

where $(a)$ is due to the invertibility of $\boldsymbol{h}$.

Because of the continuity of $\boldsymbol{h}$, the set $\boldsymbol{h}(\mathcal{C}_1)$ is a closed connected set containing $\boldsymbol{h}(\bar{\boldsymbol{x}})$. However, $\boldsymbol{h}(\mathcal{C}_1) \cap (\mathcal{X} \backslash \mathrm{cl}(\mathcal{E}_{\boldsymbol{h}})) = \phi$. This means that

$$\boldsymbol{h}(\mathcal{C}_1) \subseteq \mathcal{C}_2, \tag{12}$$

otherwise $\boldsymbol{h}(\mathcal{C}_1)$ would be disconnected.

Note that $\mathcal{C}_1$ and $\mathcal{C}_2$ are closed, connected and disjoint sets (by the property of connected components). Therefore, one can define $\epsilon$ as follows:

$$\epsilon := \mathrm{dist}(\mathcal{C}_1, \mathcal{C}_2) > 0. \tag{13}$$

Now, take any point $\boldsymbol{x}_b \in \mathrm{bd}(\mathcal{C}_1)$, where $\mathrm{bd}(\mathcal{C}_1)$ denotes the boundary of $\mathcal{C}_1$. Due to the continuity of $\boldsymbol{h}$, there exists a $\delta > 0$ such that

$$\boldsymbol{h}(\mathcal{N}_{\delta}(\boldsymbol{x}_b)) \subseteq \mathcal{N}_{\epsilon/4}(\boldsymbol{h}(\boldsymbol{x}_b)). \tag{14}$$

However, take any point $\boldsymbol{z} \in \mathcal{N}_{\delta}(\boldsymbol{x}_b) \backslash \mathrm{cl}(\mathcal{E}_{\boldsymbol{h}})$ with $\|\boldsymbol{z} - \boldsymbol{x}_b\|_2 < \epsilon/4$. Such a point exists because any neighborhood of a point on the boundary of a closed set has a non-empty intersection with the complement of the closed set. Therefore, we have

$$\boldsymbol{h}(\boldsymbol{z}) = \boldsymbol{z} \text{ because } \boldsymbol{z} \notin \mathcal{E}_h. \tag{15}$$

Since (12) implies that $\boldsymbol{h}(\boldsymbol{x}_b) \in \mathcal{C}_2$,

$$\|\boldsymbol{h}(\boldsymbol{x}_b) - \boldsymbol{x}_b\|_2 \geq \epsilon \quad \text{and} \quad \mathrm{dist}(\boldsymbol{x}_b, \mathcal{N}_{\epsilon/4}(\boldsymbol{h}(\boldsymbol{x}_b))) \geq \frac{3\epsilon}{4}.$$

Therefore

$$\mathrm{dist}(\boldsymbol{x}_b, \mathcal{N}_{\epsilon/4}(\boldsymbol{h}(\boldsymbol{x}_b))) \leq \|\boldsymbol{x}_b - \boldsymbol{z}\|_2 + \mathrm{dist}(\boldsymbol{z}, \mathcal{N}_{\epsilon/4}(\boldsymbol{h}(\boldsymbol{x}_b)))$$
$$\implies \frac{3\epsilon}{4} \leq \frac{\epsilon}{4} + \mathrm{dist}(\boldsymbol{z}, \mathcal{N}_{\epsilon/4}(\boldsymbol{h}(\boldsymbol{x}_b)))$$
$$\implies \mathrm{dist}(\boldsymbol{z}, \mathcal{N}_{\epsilon/4}(\boldsymbol{h}(\boldsymbol{x}_b))) \geq \frac{\epsilon}{2}$$
$$\implies \mathrm{dist}(\boldsymbol{h}(\boldsymbol{z}), \mathcal{N}_{\epsilon/4}(\boldsymbol{h}(\boldsymbol{x}_b))) \geq \frac{\epsilon}{2}, \tag{16}$$

where (16) is by (15). Note that (16) is a contradiction to (14). Hence, we have

$$\boldsymbol{x} \in \mathcal{C} \text{ for any } \mathcal{C} \in \mathrm{conn}(\mathrm{cl}(\mathcal{E}_{\boldsymbol{h}})) \implies \boldsymbol{h}(\boldsymbol{x}) \in \mathcal{C}.$$

This concludes the proof. $\qquad\square$

**Lemma A.3.** *Let $\boldsymbol{g}^{\star}$ be L-Lipschitz continuous. Suppose that any pair of $(\mathbb{P}_{\boldsymbol{x}|u_i}, \mathbb{P}_{\boldsymbol{x}|u_j})$ satisfies the r-SDC (cf. Definition 5). Then*

$$\|\widehat{\boldsymbol{g}}(\boldsymbol{x}) - \boldsymbol{g}^{\star}(\boldsymbol{x})\|_2 \leq 2rL. \tag{17}$$

**Proof Idea:** The proof is by contradiction. Suppose that under the conditions of Lemma A.3, Eq. (17) does not hold for some $\overline{\boldsymbol{x}} \in \mathcal{X}$. Then, there would exist a continuous non-trivial admissible MPA $\overline{\boldsymbol{h}}$ of $\{\mathbb{P}_{\boldsymbol{x}|u=u_i}\}_{i=1}^{I}$ such that $\|\overline{\boldsymbol{h}}(\overline{\boldsymbol{x}}) - \overline{\boldsymbol{x}}\|_2 > 2r$. However, this would imply, using Lemma A.2, that one can construct an open, connected, disjoint set pair $(\mathcal{A}, \mathcal{B})$ whose diameters are large, which is a contradiction to $r$-SDC that $M \leq r$.

*Proof of Lemma A.3.* Suppose that there exists $\overline{\boldsymbol{x}} \in \mathcal{X}$ such that

$$\|\widehat{\boldsymbol{g}}(\overline{\boldsymbol{x}}) - \boldsymbol{g}^{\star}(\overline{\boldsymbol{x}})\|_2 > 2rL. \tag{18}$$

Eq. (18) means that $\widehat{\boldsymbol{g}} \neq \boldsymbol{g}^{\star}$. By Lemma 1, we have that

$$
\begin{aligned}
& \widehat{\boldsymbol{g}}_{\#}\mathbb{P}_{\boldsymbol{x}|u=u_i} = \mathbb{P}_{\boldsymbol{y}|u=u_i} \\
\Longleftrightarrow\ & \widehat{\boldsymbol{g}}_{\#}\mathbb{P}_{\boldsymbol{x}|u=u_i} = \boldsymbol{g}^{\star}_{\#}\mathbb{P}_{\boldsymbol{x}|u=u_i} \\
\Longleftrightarrow\ & \boldsymbol{g}^{\star-1} \circ \widehat{\boldsymbol{g}}_{\#}\mathbb{P}_{\boldsymbol{x}|u=u_i} = \boldsymbol{g}^{\star-1} \circ \boldsymbol{g}^{\star}_{\#}\mathbb{P}_{\boldsymbol{x}|u=u_i} \\
\Longleftrightarrow\ & \boldsymbol{g}^{\star-1} \circ \widehat{\boldsymbol{g}}_{\#}\mathbb{P}_{\boldsymbol{x}|u=u_i} = \mathbb{P}_{\boldsymbol{x}|u=u_i}
\end{aligned}
$$

As $\widehat{\boldsymbol{g}} \neq \boldsymbol{g}^{\star}$, the function $\overline{\boldsymbol{h}} := \boldsymbol{g}^{\star-1} \circ \widehat{\boldsymbol{g}} \neq \mathbb{I}$ is a continuous admissible MPA of $\{\mathbb{P}_{\boldsymbol{x}|u=u_i}\}_{i=1}^{I}$. This implies that

$$\widehat{\boldsymbol{g}} = \boldsymbol{g}^{\star} \circ \overline{\boldsymbol{h}}. \tag{19}$$

Using (19), Eq. (18) implies that

$$
\begin{aligned}
\|\boldsymbol{g}^{\star} \circ \overline{\boldsymbol{h}}(\overline{\boldsymbol{x}}) - \boldsymbol{g}^{\star}(\overline{\boldsymbol{x}})\|_2 &> 2rL \\
\Longrightarrow\ L\|\overline{\boldsymbol{h}}(\overline{\boldsymbol{x}}) - \overline{\boldsymbol{x}}\|_2 &> 2rL \\
\Longrightarrow\ \|\overline{\boldsymbol{h}}(\overline{\boldsymbol{x}}) - \overline{\boldsymbol{x}}\|_2 &> 2r.
\end{aligned} \tag{20}
$$

Note that one can re-express (20) as

$$\|\overline{\boldsymbol{h}}(\overline{\boldsymbol{x}}) - \overline{\boldsymbol{x}}\|_2 = 2r + \epsilon, \tag{21}$$

using a certain $\epsilon > 0$. By Lemma A.2, we know that

$$\overline{\boldsymbol{x}} \in \overline{\mathcal{C}}, \quad \overline{\boldsymbol{h}}(\overline{\boldsymbol{x}}) \in \overline{\mathcal{C}},$$

where $\overline{\mathcal{C}}$ is a connected component of $\mathrm{cl}(\mathcal{E}_{\overline{\boldsymbol{h}}})$.

Now, let $d > 0$ and

$$\mathcal{T}_d = \mathcal{N}_d(\overline{\boldsymbol{x}}) \cap \overline{\mathcal{C}}.$$

Let $\mathcal{R}_d$ denote the connected component of $\mathcal{T}_d$ that contains $\overline{\boldsymbol{x}}$. Note that we need to consider the connected component as $\mathcal{T}_d$ can be a disconnected set. An illustration of these sets can be seen in Fig. 12.

From (21), it is easy to see that $\mathrm{dia}(\overline{\mathcal{C}}) \geq 2r + \epsilon$. Then, when $d \leq 2r + \epsilon$, since $\overline{\mathcal{C}}$ is connected, $\mathrm{bd}(\mathcal{N}_d(\overline{\boldsymbol{x}})) \cap \overline{\mathcal{C}} \neq \phi$. Note that the connected property $\overline{\mathcal{C}}$ is necessary for $\mathrm{bd}(\mathcal{N}_d(\overline{\boldsymbol{x}})) \cap \overline{\mathcal{C}} \neq \phi$ to hold for any $d \leq 2r + \epsilon$. One can further select $\boldsymbol{w} \in \mathrm{bd}(\mathcal{N}_d(\overline{\boldsymbol{x}})) \cap \overline{\mathcal{C}}$ such that $\boldsymbol{w} \in \mathrm{cl}(\mathcal{R}_d)$, i.e., $\boldsymbol{w}$ lies in the same connected component of $\mathcal{T}_d$ as $\overline{\boldsymbol{x}}$.

Note that such a $\boldsymbol{w}$ has to exist. Suppose that such a $\boldsymbol{w}$ does not exist. Then, $\mathrm{cl}(\mathcal{R}_d) \cap \mathrm{bd}(\mathcal{N}_d) = \phi$, which means that $\mathcal{R}_d$ would be disconnected from $\overline{\mathcal{C}} \backslash \mathcal{N}_d$. By the definition of $\mathcal{R}_d$, $\mathcal{R}_d$ is then disconnected from $\mathcal{T}_d \backslash \mathcal{R}_d$, which implies that $\overline{\mathcal{C}}$—that is a union of $\mathcal{R}_d$, $\mathcal{T}_d \backslash \mathcal{R}_d$, and $\overline{\mathcal{C}} \backslash \mathcal{N}_d$—is disconnected. This is a contradiction. Hence $\mathrm{cl}(\mathcal{R}_d) \cap \mathrm{bd}(\mathcal{N}_d) \neq \phi$ holds.

One can see that

$$\|\boldsymbol{w} - \overline{\boldsymbol{x}}\|_2 = d, \text{ and } \boldsymbol{w}, \boldsymbol{x} \in \mathrm{cl}(\mathcal{R}_d) \implies \mathrm{dia}(\mathcal{R}_d) \geq d, \text{ for } d \leq 2r + \epsilon.$$

Hence, there exists a large enough $d \leq 2r + \epsilon$ such that

$$0 < \mathrm{dist}(\mathcal{R}_d, \overline{\boldsymbol{h}}(\mathcal{R}_d)) < \epsilon/3.$$

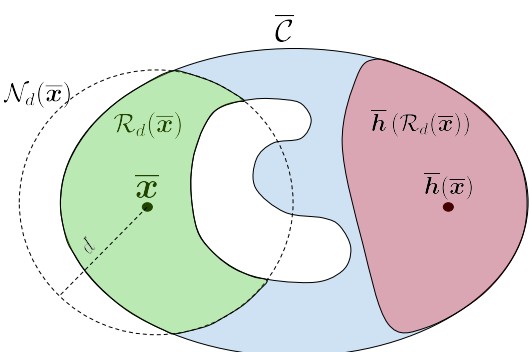

Figure 12: Illustration of the idea in the proof of Lemma A.3. The green shaded region denote $\mathcal{R}_d$. Note that $\mathcal{T}_d = \mathcal{N}_d(\overline{\boldsymbol{x}}) \cap \overline{\mathcal{C}}$ is disconnected in this case.

This implies that

$$\max\{\operatorname{dia}(\mathcal{R}_d), \operatorname{dia}(\overline{\boldsymbol{h}}(\mathcal{R}_d))\} \geq r + \epsilon/3. \tag{22}$$

Indeed, suppose that $\max(\operatorname{dia}(\mathcal{R}_d), \operatorname{dia}(\overline{\boldsymbol{h}}(\mathcal{R}_d))) < r + \epsilon/3$. Then, since $\overline{\boldsymbol{x}} \in \mathcal{R}_d$ and $\overline{\boldsymbol{h}}(\overline{\boldsymbol{x}}) \in \overline{\boldsymbol{h}}(\mathcal{R}_d)$,

$$\|\overline{\boldsymbol{x}} - \overline{\boldsymbol{h}}(\overline{\boldsymbol{x}})\|_2 \leq 2\max\{\operatorname{dia}(\mathcal{R}_d), \operatorname{dia}(\overline{\boldsymbol{h}}(\mathcal{R}_d))\} + \operatorname{dist}(\mathcal{R}_d, \overline{\boldsymbol{h}}(\mathcal{R}_d))$$
$$2r + \epsilon < 2r + 2\epsilon/3 + \epsilon/3$$
$$2r + \epsilon < 2r + \epsilon,$$

which is a contradiction. Hence,

$$\max\{\operatorname{dia}(\mathcal{R}_d), \operatorname{dia}(\overline{\boldsymbol{h}}(\mathcal{R}_d))\} \geq r + \epsilon/3. \tag{23}$$

Fig. 12 provides a simple illustration of the sets. It follow from the continuity and invertibility of $\overline{\boldsymbol{h}}$ that $\operatorname{dia}(\mathcal{R}_d) = \operatorname{dia}(\operatorname{int}(\mathcal{R}_d))$ and $\operatorname{dia}(\overline{\boldsymbol{h}}(\mathcal{R}_d)) = \operatorname{dia}(\operatorname{int}(\overline{\boldsymbol{h}}(\mathcal{R}_d)))$.

By the same argument of reaching (10), $\{\operatorname{int}(\mathcal{R}_d), \overline{\boldsymbol{h}}(\operatorname{int}(\mathcal{R}_d))\}$ forms a pair of open, connected, disjoint sets such that

$$\mathbb{P}_{\boldsymbol{x}|u_i}[\operatorname{int}(\mathcal{R}_d)] = \mathbb{P}_{\boldsymbol{x}|u_i}[\overline{\boldsymbol{h}}(\operatorname{int}(\mathcal{R}_d))]. \tag{24}$$

Note that (24) and (23) constitute a contradiction to the assumption that

$$M = \max_{(\mathcal{A},\mathcal{B})\in\mathcal{V}} \max\{\operatorname{dia}(\mathcal{A}), \operatorname{dia}(\mathcal{B})\} \leq r$$

for any open, connected, and disjoint sets $\mathcal{A}$ and $\mathcal{B}^2$

Hence, we must have

$$\|\widehat{\boldsymbol{g}}(\boldsymbol{x}) - \boldsymbol{g}^\star(\boldsymbol{x})\|_2 \leq 2rL, \forall \boldsymbol{x} \in \mathcal{X}.$$

This concludes the proof. $\square$

*Proof of Theorem 2.* We can bound the probability with which (17) does not hold as follows:

$$\Pr[(17) \text{ does not hold}]$$
$$= \Pr[M > r]$$
$$\overset{(a)}{\leq} \Pr\left[\bigcap_{i,j\in[I],i<j} M_{i,j}\right]$$
$$\overset{(b)}{=} \bigcap_{i,j\in[I],i<j} \Pr[M_{i,j}]$$
$$\leq \gamma^{\binom{I}{2}},$$

---

[2] Note that such requirements are used to ensure that the sets have nonzero measure. The statements can be simplified by replacing the "open and non-empty" sets in Definition 4 and Assumption 5 with "measurable sets with positive measures".

where $(a)$ follows because $M > r$ implies that $M_{i,j} > r, \forall i \neq j$, and $i, j \in [I]$ holds, and $(b)$ follows from the independence of the events $M_{i,j} > r$.

Hence, with probability at least $1 - \gamma^{\binom{I}{2}}$,

$$\|\widehat{\boldsymbol{g}}(\boldsymbol{x}) - \boldsymbol{g}^\star(\boldsymbol{x})\|_2 \leq 2rL, \forall \boldsymbol{x} \in \mathcal{X}.$$

The same result follows for $\widehat{\boldsymbol{f}}$ if $\boldsymbol{f}^\star$ is $L$-Lipschitz continuous, following the same procedure as above. $\qquad\square$

## D    ADDITIONAL REMARK: RELATION TO SUPERVISED DOMAIN TRANSLATION

A remark on objective (7) is that supervised domain translation can be seen as a special case of (7). When paired samples $\{\boldsymbol{x}_i, \boldsymbol{y}_i\}_{i=1}^N$ are available, one can view the auxiliary information $u$ as the sample identity. Specifically, $\mathbb{P}_{\boldsymbol{x}|u=u_i}$ and $\mathbb{P}_{\boldsymbol{y}|u=u_i}$ are Dirac delta distributions peaked at $\boldsymbol{x}_i$ and $\boldsymbol{y}_i$, respectively. Matching distributions between $\mathbb{P}_{\boldsymbol{x}|u=u_i}$ and $\boldsymbol{f}_{\#\mathbb{P}_{\boldsymbol{y}|u=u_i}}$ will be equivalent to enforcing $\boldsymbol{x}_i = \boldsymbol{f}(\boldsymbol{y}_i)$. Therefore, the sample loss will be equivalent to minimizing the following objective:

$$\underset{\boldsymbol{f}, \boldsymbol{g}}{\text{minimize}} \sum_{i=1}^N \|\boldsymbol{x}_i - \boldsymbol{f}(\boldsymbol{y}_i)\|_2^2 + \|\boldsymbol{y}_i - \boldsymbol{g}(\boldsymbol{x}_i)\|_2^2,$$

which is exactly the supervised learning loss. This makes the distribution matching problem boil down to a sample matching problem.

## E    SYNTHETIC DATA EXPERIMENTS

In this section, we use controlled generation to validate our identifiability theorems.

**Data Generation.** We generate $\boldsymbol{x}$ from a Gaussian mixture with $Q$ components. Let $\{\mathbb{P}_{\boldsymbol{x}}^{(q)}\}_{q=1}^Q$ denote the $Q$ component distributions of the Gaussian mixture, i.e.,

$$\mathbb{P}_{\boldsymbol{x}}^{(q)} \sim \mathcal{N}(\boldsymbol{\mu}_q, \boldsymbol{\Sigma}), \ q = 1, \dots, Q.$$

Here, each $\boldsymbol{\mu}_q$ is sampled randomly from the uniform distribution in $\mathbb{R}^2$, i.e., $\text{Unif}\left([-1, 1]^2\right)$. We set the covariance to be $\boldsymbol{\Sigma} = 0.3^2\boldsymbol{Q}$. To represent $\boldsymbol{g}^\star$, we use a three-layer multi-layer perceptron (MLP) with smoothed leaky ReLU, which is defined as $s(x) = \alpha x + (1 - \alpha)\log(1 + \exp(x))$, where we set $\alpha$ to 0.2. To make $\boldsymbol{g}^\star$ invertible, we generate the neural network weights using the same process as in (Hyvärinen & Pajunen, 1999; Zimmermann et al., 2021). Specifically, we use two-hidden units in each layer. We first generate $10{,}000$ $2 \times 2$ matrices, whose elements are sampled randomly from uniform distribution $\text{Unif}([-1, 1])$. The matrices' columns are normalized by their respective $\ell_2$ norms. In addition, only the top 25% well-conditioned matrices in terms of the condition number are used. This way, all the layers of the $\boldsymbol{g}^\star$ are relatively well-conditioned invertible matrices. Combining with the fact that the activation functions are invertible, such constructed $\boldsymbol{g}^\star$ in each trial is also invertible.

We use $N = 20{,}000$ samples in both domains, denoted as $\{\boldsymbol{x}_n\}_{n=1}^N, \{\boldsymbol{y}_n\}_{n=1}^N$, to be the training samples. In addition, we have 1,000 testing samples. The data generation process is as follows:

$$\boldsymbol{\mu}_q \sim \text{Unif}([-1, 1]^2), \ \forall q \in [Q],$$
$$\boldsymbol{x}_{(q-1)N_q+n} \sim \mathbb{P}_{\boldsymbol{x}}^{(q)}, \ \forall n \in [N_q] \ \forall q \in [Q],$$
$$\boldsymbol{y}_{(q-1)N_q+n} = \boldsymbol{g}^\star(\boldsymbol{x}_{(q-1)N_q+n}),$$

where $N_q = \lfloor 20000/Q \rfloor$, indicating that the mixture components have equal probability. In our experiments we use $(\boldsymbol{x}_n, \boldsymbol{y}_n)$'s association with one of the $Q$ mixture components as our auxiliary variable. Therefore, we have $I = Q$. In addition, $u$ is uniformly distributed, i.e., $\Pr(u = u_q) = 1/Q, \forall q \in [Q]$, and

$$\mathbb{P}_{\boldsymbol{x}|u=u_q} = \mathbb{P}_{\boldsymbol{x}}^{(q)}.$$

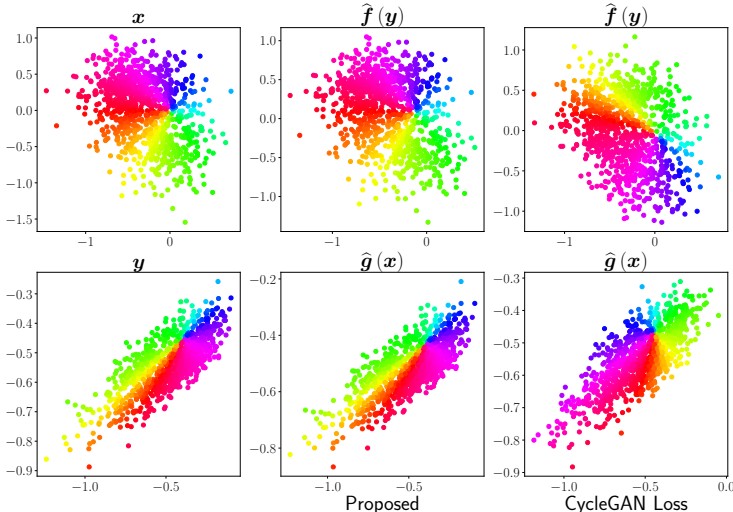

Figure 13: Scatter plots of the source and translated samples. The proposed method uses $I = Q = 3$.

**Evaluation Metric.** In the synthetic data, we have access to the ground-truth pairs $(\boldsymbol{x}_n, \boldsymbol{y}_n)$. Hence, we measure the translation error (TE) using

$$\mathrm{TE} = \sum_{n=1}^{N} \mathrm{1}/\mathrm{2}N \left( \|\widehat{\boldsymbol{g}}(\boldsymbol{x}_n) - \boldsymbol{y}_n\|_2^2 + \|\widehat{\boldsymbol{f}}(\boldsymbol{y}_n) - \boldsymbol{x}_n\|_2^2 \right).$$

**Implementation Details.** To represent $\boldsymbol{g} : \mathbb{R}^2 \to \mathbb{R}^2$ and $\boldsymbol{f} : \mathbb{R}^2 \to \mathbb{R}^2$, we use three-layer MLPs, where 256 hidden units are used in each of the 2 hidden layers. We also use leaky ReLU activations with a slope of 0.2. The discriminator is a five-layer MLP with 128 hidden units in each of the hidden layers. Each layer, except for the last, is followed by layer normalization (Ba et al., 2016) and leaky ReLU activations (Maas et al., 2013) with a slope of 0.2. We use the same architecture for all $I$ discriminators in DIMENSION. In the synthetic-data experiments, we implement the distribution matching module using the least-square GAN loss (Mao et al., 2017).

**Baseline.** In the sythetic experiments, our purpose is to show the lack of translation identifiability of naive distribution matching. Hence, we use the CycleGAN loss in (3) as a benchmark.

**Hyperparameter Settings.** We use the Adam optimizer with an initial learning rate of 0.0001 with hyperparameters $\beta_1 = 0.5$ and $\beta_2 = 0.999$ (Kingma & Ba, 2015). Note that $\beta_1$ and $\beta_2$ are hyperparameters of Adam that control the exponential decay rates of first and second order moments, respectively. We use a batch size of 1000 and train the models for 2000 iterations, where one iteration refers to one step of gradient descent of the translation and discriminator neural networks. We use $\lambda = 10$ for (7).

**Results.** Fig. 13 shows the scatter plots of the original and translated samples for the 1000 testing samples. Here, we set $I = Q = 3$. The original data $\{\boldsymbol{x}_n\}_{n=1}^N$ and $\{\boldsymbol{y}_n\}_{n=1}^N$ are plotted on the leftmost column. The result of translation using DIMENSION and CycleGAN Loss are presented in the middle and right columns, respectively. In order to qualitatively evaluate the translation performance, we use the same color to plot the paired data points $(\boldsymbol{x}_n, \boldsymbol{y}_n)$ and their translations $(\widehat{\boldsymbol{f}}(\boldsymbol{y}_n), \widehat{\boldsymbol{g}}(\boldsymbol{x}_n))$. The color is determined by the angle of $\boldsymbol{x}_n$ in polar coordinates.

As one can see, the supports of $\boldsymbol{x}$ and $\widehat{\boldsymbol{f}}(\boldsymbol{y})$ (as well as those of $\boldsymbol{y}$ and $\widehat{\boldsymbol{g}}(\boldsymbol{x})$) are well matched by both methods. This implies that both methods can match the distributions fairly well. However, CycleGAN Loss misaligns the samples (by observing the color). The results given by DIMENSION does not have this misalignment issue.

Fig. 14 shows the average TE (over 10 random trials) and the standard deviation attained by DIMENSION and the baseline under different $Q$'s. Here, we also set $I = Q$ as before. One can see that the average TE decreases with the increase in $I$. Notably, the *variance* of TE also becomes

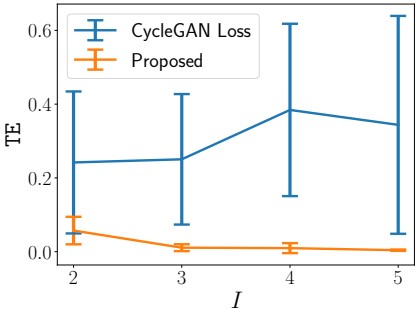

Figure 14: TE under various $I$'s.

much smaller when $I$ grows from 2 to 5—this shows more stable translation performance when $I$ increases. The result is consistent with our theorems, which shows that having a larger $I$ has a better chance to avoid MPA.

## F  REAL-DATA EXPERIMENT SETTING AND ADDITIONAL RESULTS

In this section we provide details of the real-data experiment settings.

### F.1  OBTAINING $\{u_1, \ldots, u_I\}$

**MrM and ErS Datasets.** For these two datasets, we use the available category labels as the alphabet of $u$. Specifically, for MrM dataset, we use $u \in \{1, \ldots, 10\}$, i.e., the labels of the identity of digits. For ErS datasets, we use $u \in \{\text{shoes}, \text{sandals}, \text{slippers}, \text{boots}\}$, which indicates the types of the shoes/edges.

**CB Dataset.** In this dataset, we designate the alphabet of $u$ to be $u_1 = ''$ black hair$''$, $u_2 = ''$ non-black hair$''$, $u_3 = ''$ male$''$, $u_4 = ''$ female$''$. This information is not fully available in the original CB dataset (to be specific, Bitmoji (Mozafari, 2020) has the gender attributes available but the hair color is not available). We use the foundation model, namely, CLIP (Radford et al., 2021), to acquire the hair color information of each Bitmoji face. Specifically, we use the text prompts "a cartoon of a person whose hair color is mostly black" and "a cartoon of a person whose hair color is not black". The presence of black hair for each image is decided based on cosine distance of the image embedding with the text embeddings of the two prompts.

### F.2  NEURAL NETWORK DETAILS

We use the nomenclature in Table 2 to describe the neural network architecture. For example,

$$\text{Conv-(C-}N_{\text{in}}\text{ - }N_{\text{out}}\text{, K-}N_k\text{, S-}N_s\text{, ZP-}N_p\text{), LN, LeakyReLU}$$

refers to a convolutional layer with $N_{\text{in}}$ input channels and $N_{\text{out}}$ output channels; K-$N_k$ means that the size of kernel is $N_k$; S-$N_s$ means that the stride is $N_s$; and ZP-$N_p$ means that the zero padding has a size of $N_p$. The convolutional layer is followed by layer normalization (LN) and then LeakyReLU activations.

The translation neural networks, $\boldsymbol{g}$ and $\boldsymbol{f}$, for images of size $256 \times 256$ follow the architecture outlined in Table 3. For images of size $128 \times 128$, a modified architecture is used, where one down-sampling layer (see Layer #6) and one up-sampling layer (Layer #11) in Table 3 are not included. For images of size $32 \times 32$, three down-sampling layers (indices from #4 to #6) and three up-sampling layers (indices from #11 to #13) are not included.

ResBlock refers to block of convolutional layers with shortcut connection and optional downsampling. Specifically, ResBlock-(C-$M$-$N$, *Operation*) is composed of two smaller blocks, namely, Process-(C-$M$-$N$, *Operation*) and Shortcut-(C-$M$-$N$, *Operation*). The Process-(C-$M$-$N$, *Operation*) block has the following layers:

Table 2: Nomenclature for neural network components

| Abbreviation | Definition |
|---|---|
| Conv | Convolutional Layer |
| IN | Instance normalization |
| ReLU | ReLU activation |
| LeakyReLU | Leaky-ReLU activation with 0.2 slope |
| Tanh | tanh activation function |
| UpSample | Upsample using nearest neighbor with scale factor of 2 |
| DownSample | Downsample using average pooling with a scale factor of 2 |
| K-$N$ | Kernel (filter) of size $N$ |
| S-$N$ | Stride of size $N$ |
| ZP-$N$ | Zero Padding of size $N$ |
| C-$M$-$N$ | $M$ input and $N$ output channels |

Table 3: Translation neural network architecture for $\boldsymbol{f}$ and $\boldsymbol{g}$.

| Layer Number | Layer Details |
|---|---|
| 1 | Conv-(C-3-64, K-1, S-1, ZP-0) |
| 2 | ResBlock-(C-64-128 , DownSample) |
| 3 | ResBlock-(C-128-256, DownSample) |
| 4 | ResBlock-(C-256-512, DownSample) |
| 5 | ResBlock-(C-512-512, DownSample) |
| 6 | ResBlock-(C-512-512, DownSample) |
| 7 | ResBlock-(C-512-512, –) |
| 8 | ResBlock-(C-512-512, –) |
| 9 | ResBlock-(C-512-512, –) |
| 10 | ResBlock-(C-512-512, –) |
| 11 | ResBlock-(C-512-512, UpSample) |
| 12 | ResBlock-(C-512-512, UpSample) |
| 13 | ResBlock-(C-512-256, UpSample) |
| 14 | ResBlock-(C-256-128, UpSample) |
| 15 | ResBlock-(C-128-64 , UpSample) |
| 16 | Conv-(C-64-3, K-1, S-1, ZP-0) |

1. IN, LeakyReLU, Conv-(C-$M$-$M$, K-3, S-1, ZP-1)
2. *Operation*
3. IN, LeakyReLU, Conv-(C-$M$-$N$, K-3, S-1, ZP-1)

The Shortcut-(C-$M$-$N$, *Operation*) block consists of the following layers:

1. Conv-(C-$M$-$N$, K-1,S-1,ZP-0)
2. *Operation*

Let $\boldsymbol{z}$ denote the input to the ResBlock and $\boldsymbol{w}$ the output of the ResBlock. Then the forward pass of ResBlock is expressed as follows:

$$\boldsymbol{w} = \text{ResBlock}(\boldsymbol{z}) = \text{Process}(\boldsymbol{z}) + \text{Shortcut}(\boldsymbol{z}).$$

We use multi-task discriminators (Liu et al., 2019) with output dimension of $I$ to represent $\boldsymbol{d}_x^{(i)}, \boldsymbol{d}_y^{(i)}, \forall i \in [I]$. Specifically, each of the multi-task discriminators $\boldsymbol{d}_x$ and $\boldsymbol{d}_y$ has $I$ output dimensions. The $i$th outputs of $\boldsymbol{d}_x$ and $\boldsymbol{d}_y$ correspond to $\boldsymbol{d}_x^{(i)}$ and $\boldsymbol{d}_y^{(i)}$, respectively.

Table 4: Discriminator architecture for $\boldsymbol{d}_x : \mathbb{R}^{256 \times 256 \times 3} \to \mathbb{R}^I$, $\boldsymbol{d}_y : \mathbb{R}^{256 \times 256 \times 3} \to \mathbb{R}^I$.

| Layer Number | Layer Details |
|:---:|:---:|
| 1 | Conv-(C-3-64, K-1, S-1, ZP-0) |
| 2 | ResBlock-(C-64-128, DownSample), |
| 3 | ResBlock-(C-128-256, DownSample), |
| 4 | ResBlock-(C-256-512, DownSample), |
| 5 | ResBlock-(C-512-512, DownSample), |
| 6 | ResBlock-(C-512-512, DownSample), |
| 7 | ResBlock-(C-512-512, DownSample), LeakyReLU |
| 8 | Conv-(C-512,512, K-4, S=1, ZP=0), LeakyReLU |
| 9 | Reshape-512 |
| 10 | Linear-(512,I) |

### F.3 HYPERPARAMETER SETTING

We use the Adam optimizer with an initial learning rate of $0.0001$ with hyperparameters $\beta_1 = 0.0$ and $\beta_2 = 0.999$ (Kingma & Ba, 2015). Note that $\beta_1$ and $\beta_2$ are hyperparameters of Adam that control the exponential decay rates of first and second order moments, respectively. We set our regularization parameter $\lambda = 10$. We use a batch size of 16. We train the networks for 100,000 iterations. Following standard practice, we add squared $\ell_2$-norm regularization on the network parameters and use a *weight decay* of 0.00001. For the translation tasks with $256 \times 256$ images (CelebA-HQ to Bitmoji Faces), the runtime using a single Tesla V100 GPU is approximately 55 hours. For the translation tasks with $128 \times 128$ images (Edges to Rotated Shoes), the runtime using a single Tesla V100 GPU is approximately 35 hours. In order to stabilize the GAN training dynamics, we add a gradient penalty term. This term penalizes discriminators' large gradients , which is known to help the convergence of the GAN objective (Mescheder et al., 2018). We modified the rgeularizer to accommodate our diversified DT loss function. The modified regularization term is as follows:

$$\mathcal{R} = \frac{\gamma}{2} \Pr(u = u_i) \left( \mathbb{E}_{\boldsymbol{x} \sim \mathbb{P}_{\boldsymbol{x}|u=u_i}} \|\nabla \boldsymbol{d}_x^{(i)}\|_2^2 + \mathbb{E}_{\boldsymbol{y} \sim \mathbb{P}_{\boldsymbol{y}|u=u_i}} \|\nabla \boldsymbol{d}_y^{(i)}\|_2^2 \right),$$

where $\nabla \boldsymbol{d}_x^{(i)}$ denotes the gradient of $\boldsymbol{d}_x^{(i)}$. We set the value of $\gamma$ to be $1.0$. We take exponential moving average (EMA) of the parameters during training as the final estimate of the parameters of the trained neural networks. We use a weighting factor of 0.999. This has been observed to improve the performance of GANs (Karras et al., 2017; Yaz et al., 2018).

### F.4 DATASET DETAILS

**MNIST to Rotated MNIST (MrM).** We use $60,000$ training samples of the MNIST digits (LeCun et al., 2010) that have a dimension $28 \times 28$ as the $\mathcal{X}$-domain. For the †-domain, each of the $60,000$ digits is rotated by 90 degrees. The orders of samples are shuffled in both domains to "break" the content correspondence. Under this setting, each $\boldsymbol{x}$ has a ground-truth correspondence $\boldsymbol{y}$.

**Edges to Rotated Shoes (ErS).** Edges2Shoes dataset (Isola et al., 2017) consists of $49,825$ training samples. We resize the all images to have $128 \times 128$ pixels. The $\mathcal{X}$-domain corresponds to the *edges of the shoes*, and the $\mathcal{Y}$-domain corresponds to the *shoes* that are rotated by 90 degrees. Like in the MrM dataset, the ground-truth correspondence is known to us, which can assist evaluation.

**CelebA-HQ to Bitmoji Faces (CB)** We use $29,900$ training samples from CelebA-HQ (Karras et al., 2017) as the $\boldsymbol{x}$-domain, and $3,984$ training samples from Bitmoji faces (Mozafari, 2020) as the $\boldsymbol{y}$-domain. Note that Bitmoji Faces consists of only $4,084$ samples in total, of which $100$ samples are held out as the test samples. We resize all images in both domains to have $256 \times 256$ pixels. Unlike the previous two datasets, the ground-truth correspondence is *not* known to us in this dataset.

**Evaluation Details.**    The LPIPS score is computed using 100 test samples.    Pre-trained AlexNet(Krizhevsky et al., 2012; Zhang et al., 2018) is used in order to compute the LPIPS scores.

The `FID` score is computed using 1000 translated and real samples for each domain. Pre-trained Inception-v3 (Szegedy et al., 2016) is used in order to compute the `FID` scores.

## F.5 BASELINES

We use `CycleGAN+Id` (Zhu et al., 2017)[3], `UNIT` (Liu et al., 2017) [4], `MUNIT` (Huang et al., 2018) [4], `U-GAT-IT` (Kim et al., 2020) [5], `StarGAN-v2` (Choi et al., 2020) [6], `ZeroDIM` (Gabbay et al., 2021) [7], `OverLORD` (Gabbay & Hoshen, 2021) [8], `Hneg-SRC` (Jung et al., 2022) [9], `GP-UNIT` (Yang et al., 2023) [10], and the plain-vanilla `CycleGAN Loss` in (3) as the baselines.

For `StarGAN-v2` and `GP-UNIT`, training is done with their default settings (specifically, the configurations for the 'AFHQ' dataset in their papers are used). For `CycleGAN+Id`, `UNIT`, `MUNIT`, `U-GAT-IT`, and `Hneg-SRC`, we train the models for 200,000 iterations. We use a batch size of 8 for these methods except for `U-GAT-IT`, which uses 4 in order to control the computational load and runtime. These parameters are carefully set for the baselines to our best extent. For `OverLORD` (Gabbay & Hoshen, 2021), we use the setting used for male to female translation task on CelebA-HQ dataset in their paper. For `ZeroDIM` (Gabbay et al., 2021) , we use the setting used for experiments on FFHQ dataset, which has a similar size as the datasets used in our paper. Note that `ZeroDIM` also uses the same auxiliary variables as those used in the proposed method.

## F.6 ADDITIONAL RESULTS

In this subsection, we present additional qualitative and quantitative results.

Fig. 16 shows the result of translating Bitmoji faces (B) to celebrity proflie photos (C). As mentioned in the main text, translating from the B domain to the C domain is a hard task as the learned translation function needs to "fill in" a lot of details to make the generated profiles photorealistic. Visually, one can see that the proposed method (with $I = 4$) exhibits much more intuitive content alignment relative to the baselines. In addition, the proposed method using $I = 2$ (only using 'male' and 'female' as the auxiliary variable alphabet) also provides more satisfactory results relative to the baselines. This echos our theoretical claim that the chance of attaining translation identifiability grows quickly when $I$ increases. It also shows that diversifying the distributions to be matched, even if just one more distribution pair is included, helps improve the final performance.

Fig. 15 shows the result of translating edges (E) to rotated shoes (rS). Visually, our method significantly outperforms the baselines in terms of content alignment. It is interesting to notice that, although "edges to shoes" (no rotation) is a well studied dataset, our experiments show that a simple rotation makes most of the existing methods struggle to produce reasonable results. However, our method is insensitive to this kind of geometric changes. In the literature, the baselines `U-GAT-IT` (Kim et al., 2020) and `GP-UNIT` (Yang et al., 2023) were shown to be good at handling certain geometric variations. However, one can see that their performance over the ErS dataset is still far from ideal. The result shows the importance of taking transaltion identifiability into account, especially when drastic geometric changes happen across domains.

Fig. 18 and Fig. 17 show similar results for the translation of CelebA-HQ (C) to Bitmoji Faces (B) and Rotated Shoes (rS) to Edges (E), respectively. Fig. 19 shows the translations between MNIST (M) and rotated MNIST digits (rM).

---

[3]https://github.com/junyanz/pytorch-CycleGAN-and-pix2pix.git

[4]https://github.com/NVlabs/MUNIT.git

[5]https://github.com/znxlwm/UGATIT-pytorch.git

[6]https://github.com/clovaai/stargan-v2.git

[7]https://github.com/avivga/zerodim

[8]https://github.com/avivga/overlord

[9]https://github.com/jcy132/Hneg_SRC.git

[10]https://github.com/williamyang1991/GP-UNIT.git

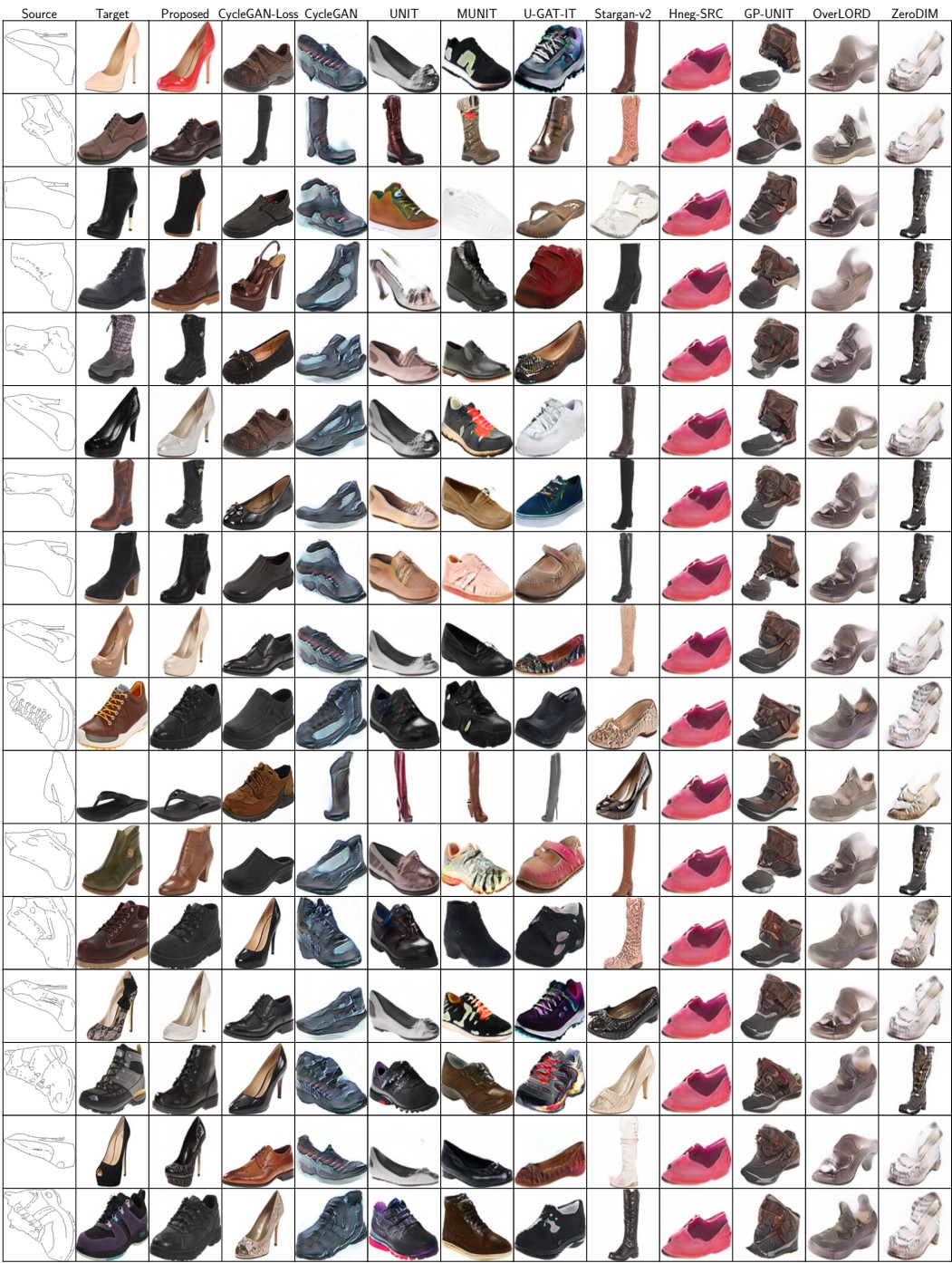

Figure 15: Translation of edges to rotated shoes. All images rotated by anit-clockwise 90 degrees for visualization.

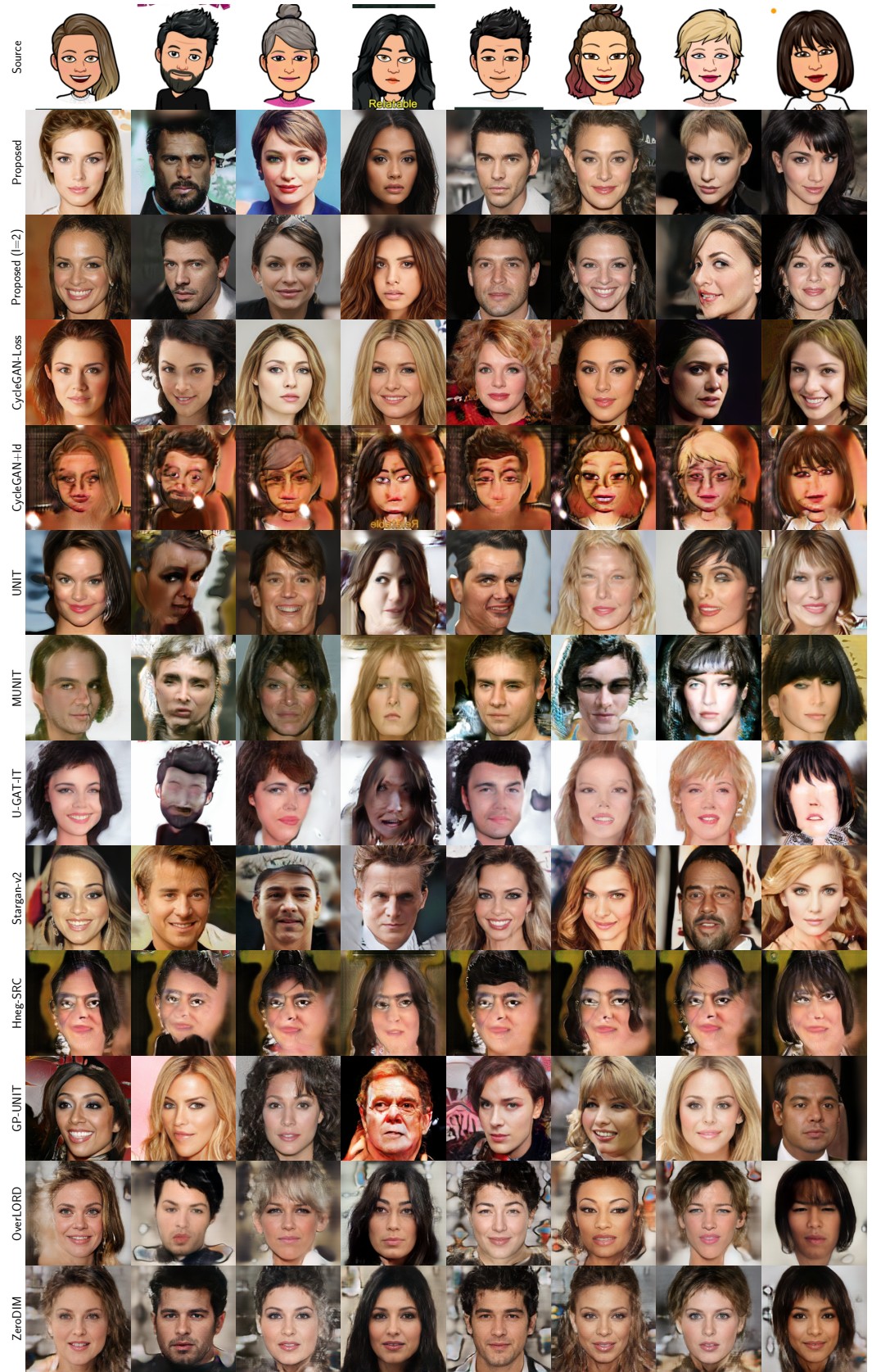

Figure 16: Translation of Bitmoji to CelebA-HQ.

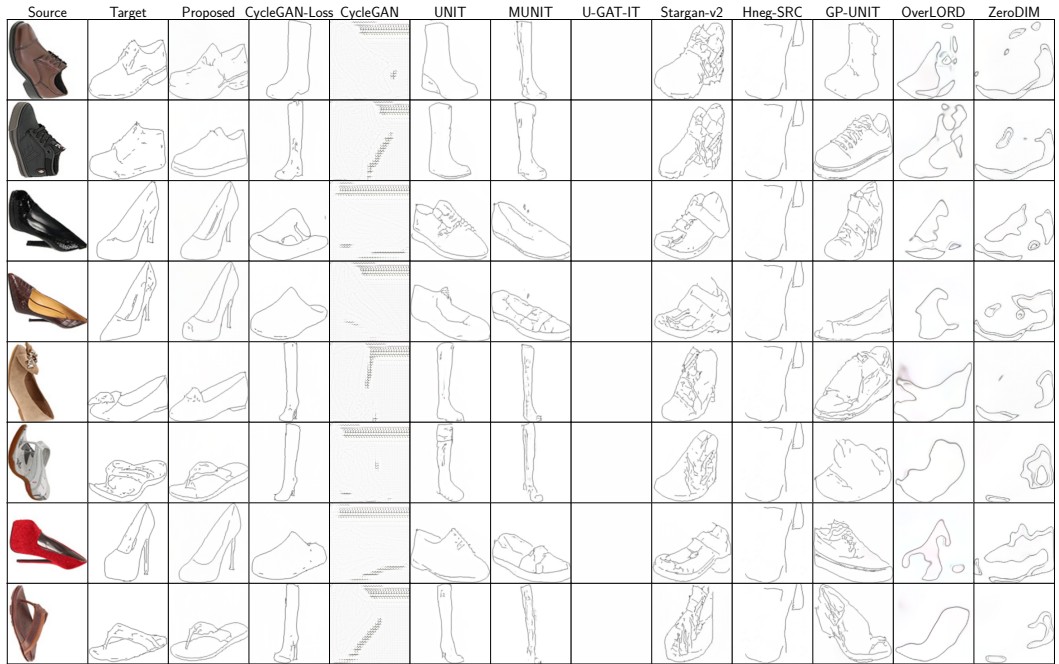

Figure 17: Translation of rotated Shoes to Edges.

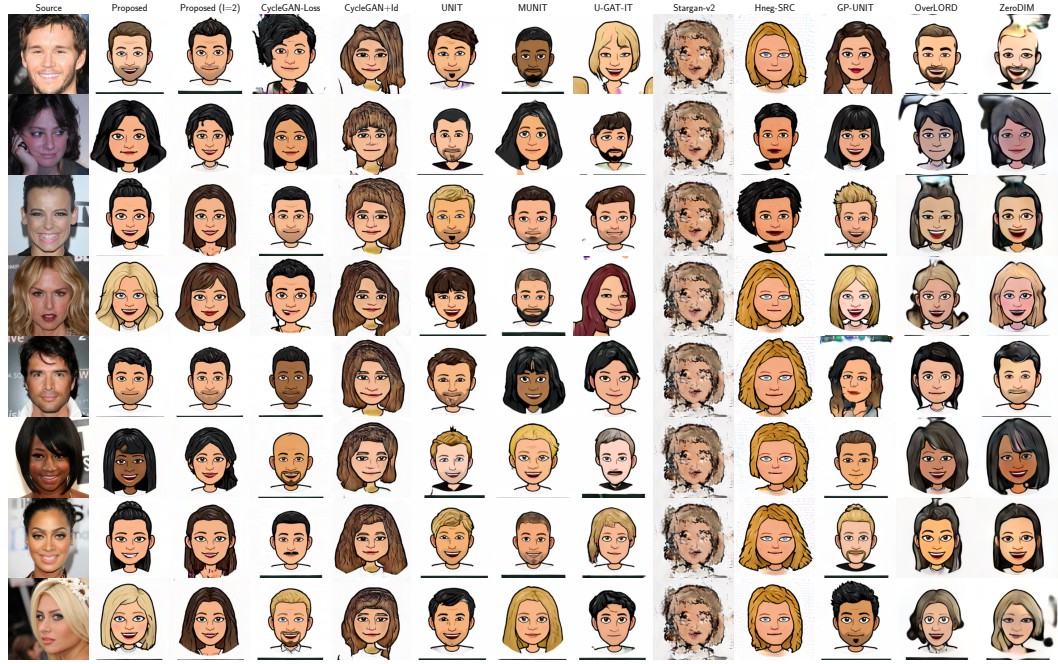

Figure 18: Translation of CelebA-HQ to Bitmoji.

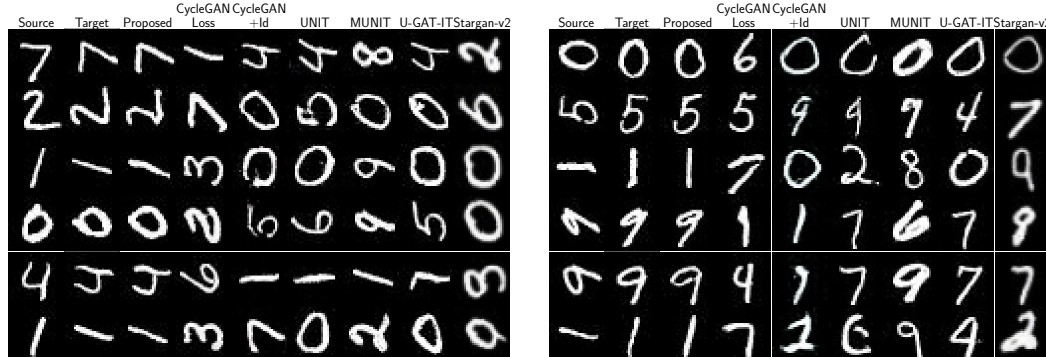

Figure 19: Translation between MNIST digits and rotated MNIST digits.

There are multiple ways to define auxiliary variables for a given UDT task. However, different choices of auxiliary information can result in different level of translation performance. For example, in the MNIST example, one can alternatively use digit shape as the alphabets of the auxiliary variable. Figure 20 shows the result of using different shapes of digits. Here we use the following attributes (with corresponding digits that has those attributes): "line" : [1,2,4,5,7], "circle" : [0,6,8,9], "curve" : [0,2,3,5,6,8,9], "vertical line" : [1, 4, 5], "horizontal line" : [2,5,4,7], "curve without loops" : [2,3,5,6,9], "only vertical line" : [1], "only horizontal line" : [2]). One can see that using digit identity as auxiliary information results in a slightly enhanced performance compared to using the digit shape as auxiliary information.

## G  IMPROVING EXISTING METHODS USING DIVERSIFIED DISTRIBUTION MATCHING

We hope to emphasize that the diversified distribution matching (DDM) principle can be combined with many other existing UDT approaches to avoid failure cases. In this section, we use our diversified distribution matching module to replace the their original ones in existing paradigms and observe the performance. For the datasets "Edges vs. Rotated Shoes" (ErS) and "CelebA-HQ vs. Bitmoji" (CB), we select the baselines that are able to generate faithful samples in the target domain based on their `FID` scores (see Table 1. To be specific, for the ErS dataset, we integrate DDM with `UNIT` (Liu et al., 2017). For the CB dataset, we combine DDM with `GP-UNIT` (Yang et al., 2023). In both cases, we keep their method-defined regularization terms and other settings unchanged. We refer to the modified methods as `UNIT-DDM` and `GP-UNIT-DDM`, respectively.

Our way of combining DDM with these existing approaches is to replace their discriminators. To obtain `UNIT-DDM` and `GP-UNIT-DDM`, we modify the discriminator neural networks of `UNIT` and `GP-UNIT` into multi-task discriminators. Specifically, for `UNIT-DDM`, the multi-scale discriminator of `UNIT` which has one output channel for each scale, is modified to produce $I$ output channels for each scale. Similarly, to obtain `GP-UNIT-DDM`, the discriminator of `GP-UNIT` is modified to have $I$ output channels instead of one output channel at the output layer. The $i$th output channel is interpreted as the $i$th discriminator associated with $u_i$.

Fig. 21 shows the qualitative results attained by the original versions of `UNIT` and `GP-UNIT` as well as their `DDM`-modified versions. One can see that there is significant improvement in terms of content alignment, without compromising the visual quality—see the `FID` and `LPIPS` scores in Table 5. This attests to the hypothesis that distribution-matching based domain translation frameworks can benefit from the proposed MPA eliminating idea.

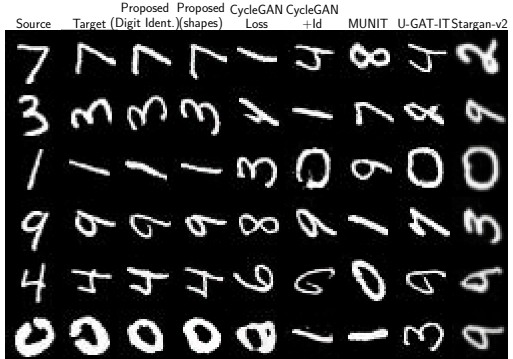

Figure 20: Result of using different auxiliary variable for MNIST digits to rotated MNIST digits task. Using shape attributes incur LPIPS=0.19 ± 0.08, compared to LPIPS=0.11 ± 0.08 for the digit identity as auxiliary variable.

Table 5: The `FID` and `LPIPS` scores attained by `UNIT` and `GP-UNIT` as well as their `DDM`-modified versions.

| Method | **FID** ($\downarrow$) | | **LPIPS** ($\downarrow$) | |
|---|---|---|---|---|
| | Edges | Shoes | Edges $\rightarrow$ Rot. Shoes | Rot. Shoes $\rightarrow$ Edges |
| UNIT | 33.95 | 96.28 | $0.49 \pm 0.035$ | $0.58 \pm 0.038$ |
| UNIT-DDM | 43.95 | 88.58 | $0.30 \pm 0.075$ | $0.35 \pm 0.092$ |

| Method | CelebA-HQ | Bitmoji |
|---|---|---|
| GP-UNIT | 32.40 | 30.30 |
| GP-UNIT-DDM | 37.79 | 30.33 |

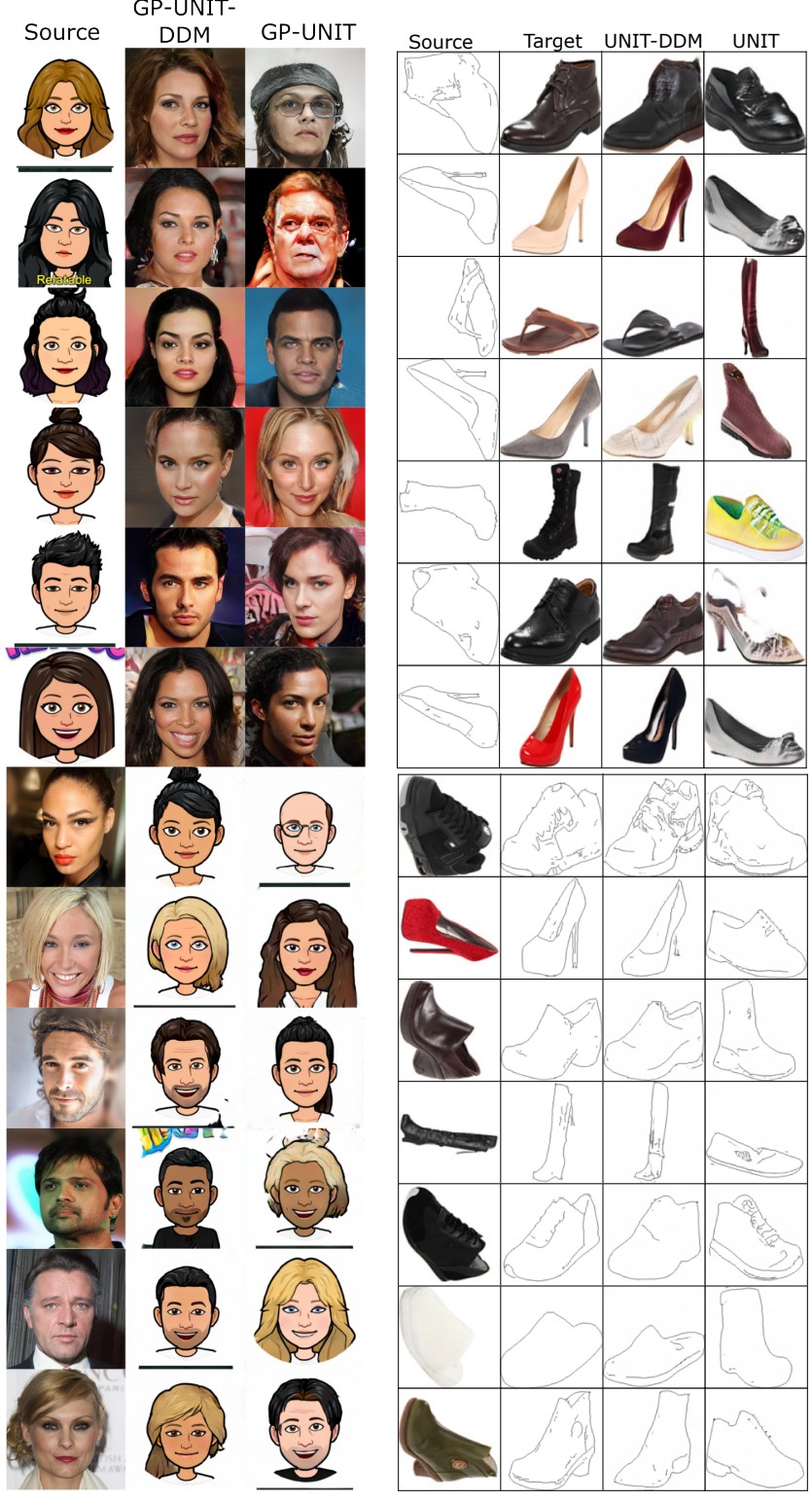

Figure 21: [Left] Result of auxiliary variable-based diverse distribution matching on GP-UNIT on Bitmoji → CelebaA-HQ translation task. [Right] Result of Conditional distribution matching on UNIT on Edges → Rotated Shoes translation task.

Table 6: LPIPS score attained by DIMENSION using random $u_i$ assignments.

| random $u_i$ proportion | MNIST → Rot. MNIST | Rot. MNIST → MNIST |
|---|---|---|
| 0% | $0.11 \pm 0.082$ | $0.09 \pm 0.047$ |
| 20% | $0.09 \pm 0.050$ | $0.08 \pm 0.040$ |
| 40% | $0.10 \pm 0.049$ | $0.13 \pm 0.064$ |
| 50% | $0.19 \pm 0.080$ | $0.19 \pm 0.083$ |
| 60% | $0.25 \pm 0.124$ | $0.21 \pm 0.086$ |

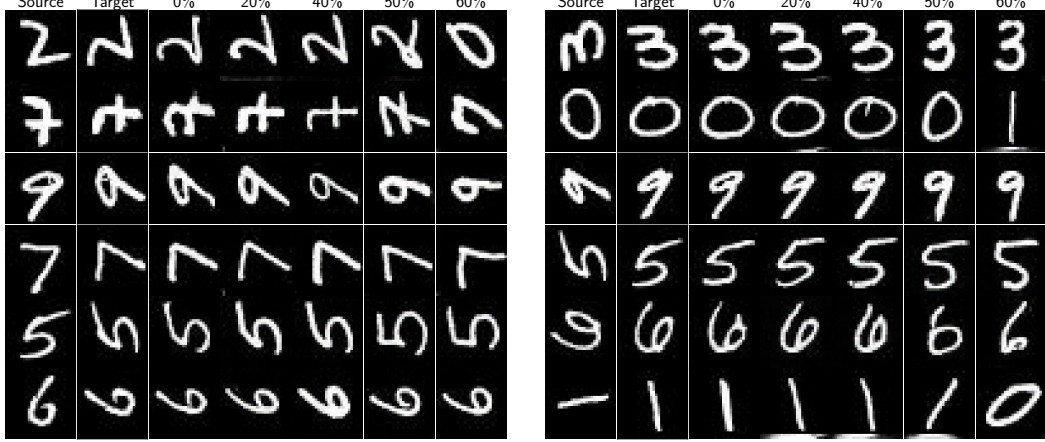

Figure 22: Result of DIMENSION under random $u_i$ assignments to various fractions of training data.

## H    ROBUSTNESS TO NOISY AUXILIARY VARIABLES.

It is of interest to know whether using noisy or wrong auxiliary variables would heavily affect the performance of DIMENSION. To this end, we assign random $u_i$'s to a fraction of the training samples in the "MNIST vs. Rotated MNIST" dataset.

Table 6 and Fig. 22 show the LPIPS scores and qualitative results attained by DIMENSION, respectively, under different fractions of random (and highly possibly wrong) auxiliary variables. Notably, there is almost no performance degradation of DIMENSION even when $40\%$ of the assigned $u_i$'s are random. This shows the method's robustness to wrong/noisy auxiliary variables.

