# OpenReview forum: "Towards Identifiable Unsupervised Domain Translation: A Diversified Distribution Matching Approach"
_ICLR.cc/2024/Conference — ICLR 2024 poster_

### Official Review · Reviewer_rKLC · 2023-10-14

**Soundness:** 3 good
**Presentation:** 3 good
**Contribution:** 2 fair
**Rating:** 6
**Confidence:** 5

**Summary:**

The paper tackles common failures in CycleGAN and variants where the desired translation functions are not successfully identified and the methods produce content-misaligned translations.

This limitation is claimed to be related to the presence of multiple translation functions (MPA). The authors introduce an MPA elimination theory and suggest a modified learning approach in which the cross-domain distributions are matched over auxiliary variable-induced subsets of the domains (e.g. translation between real human faces to cartoonized figures is conditioned on hair color and gender).

Quantitative and qualitative evaluation on several geometrically-unaligned pairs of datasets (Rotated MNIST, Rotated Edges-2-Shoes and CelebA to Bitmoji) are presented to support the theoretical claims.

**Strengths:**

The studied problem of matching the distributions of unaligned image domains is of great interest in the image-to-image translation line of work. The development of theoretical frameworks as the one introduced in this paper can shed light on the limitations of such unsupervised approaches and lead to more robust translation methods.

**Weaknesses:**

1. The experimental study is not conducted in the most relevant setting in my opinion. As the proposed method relies on several auxiliary variables (e.g. hair color in human faces or the digit class in the MNIST experiment), I believe the baselines should represent methods in weakly-supervised image-to-image translation [1]. Comparison against unsupervised methods is unfair.

2. The authors claim the auxiliary variables can be queried from available foundation models as CLIP. This idea is already explored in [1], could the authors please provide any experimental benchmark including CLIP-based annotations against [1]?

3. There are some other works relating the failures in geometrically-unaligned image domains to architectural inductive biases [2]. Moreover, methods as [2] present translations between domains with some degree of geometry variation without access to additional labels in the form proposed in this paper. Could the authors provide a comparison to [2] on the celebA-to-bitmoji?

[1] Gabbay et al. “An Image is Worth More Than a Thousand Words: Towards Disentanglement in the Wild”. In NeurIPS, 2021.

[2] Gabbay et al. “Scaling-up Disentanglement for Image Translation”. In ICCV, 2021.

**Questions:**

1. I find the qualitative results quite limited. For example, In Fig. 8, the translation from human faces to bitmoji does not preserve the facial expression. Considering that the gender and hair color is provided to the model, and the facial expression is not preserved, what other properties should the reader focus on to verify the validity of the translation?

---

> ### Author Response · Authors · 2023-11-17
>
> ## ***Response to Reviewer rKLC (Part 1)***
>
> &nbsp;
>
> **[Response to Strengths]**
>
> We thank the reviewer for highlighting the significance of our work.
>
> &nbsp;
>
> ## **Weaknesses**
> &nbsp;
> ### **Q1.**
>
> The reviewer has raised an important and relevant question. To address this concern, we have re-run some experiments using the references that the reviewer provided -- please see the updated manuscript. We thank the reviewer for providing these relevant papers to complete our numerical session.
>
> Nonetheless, please let us explain the reason why we chose the baselines in our original submission - and the reason why they are also very relevant. Please note that our identifiability analysis shows that mere distribution matching and invertibility as in objective (3) do not ensure identifiability, but using auxiliary variables can avoid content misalignment cases. To validate the above main theorem, we needed to use baselines that do not use auxiliary variables, e.g., CycleGAN and its variants. This is important, in order to demonstrate the existence of MPA and to validate our MPA-eliminating theory.
>
> That being said, we do understand and agree with the reviewer's point. It is of interest to see how the proposed solution performs against methods that use the same/similar auxiliary information. The result of the comparison is explained in the revised version as well as our response to Comment 2 below.
>
> &nbsp;
>
> ### **Q2.**
>
> Thank you for your constructive comment and pointing us to a relevant paper.  We have now included [1] as a new baseline and cited it in the place where we discuss the possibility of using CLIP to acquire auxiliary variables.
>
> Our understanding is that [1] aims at learning disentangled representations of the data, which can be applied to domain translation at test time. This is different from the UDT model in the manuscript. However, we agree with the reviewer that it is of interest to use it as a baseline as it also uses auxiliary information.
>
> For the new baseline, we use the same auxiliary variables as those used in the proposed method. To be specific, we use the 4 shoe types for Edges vs. Rotated Shoes, and hair color and gender for CelebA-HQ vs. Bitmoji. We present the result of the configuration of the baseline recommended for the FFHQ dataset in their paper, which has a similar size as the datasets used in this work. For the CelebA-HQ vs Bitmoji dataset in Table 1, Fig. 8, Fig. 16 and Fig. 18, the content alignment is indeed better than other baselines. However the translation quality is not as satisfactory. For the Edges vs. Rotated Shoes in Table 1, Fig. 8, Fig. 15 and Fig. 17, the baseline [1] still could fail to learn content aligned translation.
>
> [1] Gabbay et al. “An Image is Worth More Than a Thousand Words: Towards Disentanglement in the Wild”. In NeurIPS, 2021.
>
> &nbsp;
>
> ### **Q3.**
>
> **[New Baseline [2]]**
>
> Again, we thank the reviewer for providing the reference. In the revised version, we have included this paper as a new baseline as well.
>
> For the experiments, we used the recommended configuration of CelebA-HQ male to female task in [2]. Our observation is that the new baseline could still sometimes fail produce correct translation across the two datasets. Mainly, for the Edges vs. Rotated Shoes dataset in Table 1, Fig. 8, Fig. 15, and Fig. 17, the issue of content misalignment remains. For CelebA-HQ vs. Bitmoji in Table 1, Fig. 8, Fig. 16, and Fig. 18, the content alignment is somewhat better but the translated sample quality does not seem very satisfactory.
>
> **[Architectural inductive biases]**
>
> Another interesting point that we hope to discuss with reviewer is regarding the "architectural inductive bias" as raised in the comment.
> This also leads to an important clarification. In [2], the failure cases of existing methods on geometrically-disparate domains is attributed to the geometry preserving biases present in most of the existing methods. **We hope to clarify that the failure cases caused by "geometrical unalignment" in [2] are not the same as the "content misalignment" problem that we deal with**. In [2], the failure cases refer to the inability to generate faithful/realistic samples in the target domain (e.g., Fig. 2 in [2]). In this paper, the target problem "content misalignment" refers to failure cases such as translating "digit 7" to "rotated digit 3". There, the content (or, high-level semantic meaning) is clearly the identity of the digit and the domains are "regular display of the digits" and "rotated display of the digits". Hence, content misalignment can still exist even when the translated samples are faithful/realistic in the target domain. This is the core issue of non-identifiability problem that we aim to solve.
>
> [2] Gabbay et al. “Scaling-up Disentanglement for Image Translation”. In ICCV, 2021.

---

> > ### Comment · Reviewer_rKLC · 2023-11-17
> > **Updated score**
> >
> > I thank the authors for their detailed response. I believe the extended experimental evaluation and especially the consideration of methods that use auxiliary variables strengthens the empirical validation supporting the theoretical framework.
> >
> > I have increased my score accordingly.

---

> ### Author Response · Authors · 2023-11-17
>
> ## ***Response to Reviewer rKLC (Part 1)***
>
> &nbsp;
>
> ### **Question**
>
> The reviewer raised an interesting question. From human face to bitmoji, our sense is that there are many attributes that can verify correct content alignment. Fig. 18 in our supplementary has an enriched set of visual examples, which can  demonstrate this well.
> For example, attributes such as the beard of a person, the hair style, and the skin color can be used for visual validation of content alignment. The mouth expression is also preserved to an extent (a relatively bigger smile or a smaller smile). If one looks at the first male person in Fig. 18, the translation of our method (i.e., the first column) preserved most of the mentioned attributes across domains, including the bigger smile with teeth seen. But other methods could not do the same.
>
> &nbsp;
>
> ### **Summary**
>
> We hope our explanations and added experiments could alleviate the reviewer's concerns. Again, we hope to re-iterate that our goal is beyond only empirical success, but to validate the model postulate and the theory developed under the postulate. We acknowledge that empirical success is important (which we believe that our experiments have demonstrated). We do appreciate the recommended baselines and the push to make the numerical section more enriched. But to us, the more important point is that the experiments look consistent with our theory development, which attests to the validity of the model postulate and the identfiability analysis. These aspects often naturally lead to predictable and good numerical performance, as we saw in the original and added experiments. We hope the reviewer agrees with us that such "model hypothesis -- theory development -- experiment validation of theory" pipeline can now make good sense. In light of the above discussion, we hope the reviewer could reconsider the score.

---

### Official Review · Reviewer_siTh · 2023-10-27

**Soundness:** 3 good
**Presentation:** 4 excellent
**Contribution:** 3 good
**Rating:** 8
**Confidence:** 4

**Summary:**

The paper studies Unsupervised domain translation (UDT),
e.g. to learn to generate cartoon sketches from ID photos
without supervision. The authors study why CycleGANs fail to
learn the desired UDT function; previous work has suggested that
the reason is the existence of automorphisms of the generative
distributions; they corroborate this suggestion with a theoretical
argument and then propose a second theoretical argument that
prevents existence of such automorphisms if one introduces conditioning
on auxiliary variables. They then show the effectiveness of the proposed
automorphisms elimination approach on a few benchmarks.

**Strengths:**

1. The paper is well written and easy to read.

2. The suggested elimination idea is well-motivated and simple
  to implement.

3. While *Theorem 1* operates under idealized assumptions, *Theorem 2*
  makes an attempt to show that the author's proposal is robust
  under more realistic circumstances.

4. The UDT tasks they experiment on seem challenging enough to be interesting.

**Weaknesses:**

The abstract sounds very specialistic to me. I think the paper might be of interest to a broader audience, but some readers unfamiliar with the jargon might be put off by the abstract.

**Questions:**

My initial rating inclines towards acceptance. A limitation of my review is that I have not a direct experience with the baselines, so I cannot assess if the chosen baselines were too easy to beat.

**Questions**:
1. In assumption 1 how realistic are the invertibility assumptions?
  Are there weakened versions, e.g. in a probabilistic sense?
2. Regarding Proposition 1 and the MNIST example in Figure 1, it seems
  that for MNIST the support of $P_x$ would not be path-connected, with one path-component
  for each digit. Then Proposition 1 would not apply directly. Can you formulate a case of Proposition 1 that would apply to this case?

---

> ### Author Response · Authors · 2023-11-17
>
> ## ***Response to Reviewer siTh (Part 1)***
>
> &nbsp;
>
> Thank you for the overall positive comments.
>
> &nbsp;
>
> **[Weaknesses: Accessibility of Abstract]**
>
> Thank you for pointing out the potential weaknesses. We have revised our abstract to make it more accessible to general audience.
>
> **[Choice of Baselines]**
>
> We would like to stress that the chosen baselines are either state-of-the-art approaches or very representative and seminal works in the unsupervised *image-to-image* (I2I) translation literature. The chosen baselines are widely used as state-of-the-art baselines in unsupervised I2I literature [Jung et al., 2022; Park et al., 2020], including the most recent ones [Yang et al., 2023; Park et al., 2023; Ko et al., 2023].  Mainly, our choice of each of the baselines are motivated by the following reasons:
>
> 1. CycleGAN Loss Eq.(3) : Original CycleGAN loss [Zhu et al., 2017] without any engineering heuristics.
>     The plain-vanilla CycleGAN exactly uses the distribution matching and invertibility modules. We use it to demonstrate the failure cases when auxiliary variables are not used.
>
> 2. CycleGAN + Id [Zhu et al., 2017]: This is the performance-enhanced version of CycleGAN proposed in the same paper. The "Id" regularization can be understood as a heuristic to avoid non-identifiability.
>
> 3. UNIT [Liu et al., 2017]: A representative work on unsupervised I2I. It introduces shared latent space assumption in order to "shrink" the solution space of the CycleGAN-type loss function. Exploiting this assumption can avoid content-misalignment to a certain extent, empirically.
>
> 4. MUNIT [Huang et al., 2018]: A seminal work on multimodal unsupervised I2I. It extends the CycleGAN work to probabilistic translation. But it did not consider the theoretical issues as in our work.
>
> 5. U-GAT-IT [Kim et al., 2020]: A representative work on unsupervised I2I. The challenge that was addressed in this work is to deal with large geometric disparity between domains.
>
> 6. StarGAN-v2 [Choi et al., 2020]: One of the state-of-the-art methods on unsupervised I2I. It enabled high resolution multi-domain unsupervised I2I on human faces and animal faces.
>
> 7. Hneg-SRC [Jung et al., 2022]: One of the state-of-the-art methods in unsupervised I2I using contrastive learning based unsupervised I2I. It proposed hard negative sample mining method to improve contrastive learning regularized distribution matching framework for unsupervised I2I.
>
> 8. GP-UNIT [Yang et al., 2023]: One of the most recent state-of-the-art methods on unsupervised I2I. It leveraged pre-trained BigGAN to learn translation between domains with large structural disparity.
>
> Moreover, following suggestion from one of the reviewers, we have added the following baselines:
>
>
> 9. ZeroDIM [Gabbay et al., 2021]: A disentangled representation learning method. It also uses the same auxiliary information as the proposed method.
>
> 10. OverLORD [Gabbay and Hoshen, 2021]: A interesting non-adversarial method for learning disentangled representation. It was also applied to image-to-image translation.
>
> &nbsp;
>
> ### **Questions**
> &nbsp;
>
> **Q1.** This is a good point to discuss.
> In principle, the invertibility of translation function is a reasonable assumption if the domains can be translated back and forth. Seeking such invertible translation functions has been an important task in a vast volume of domain translation works [Zhu et al., 2017, Liu et al., 2017; Kim et all, 2020; Park et al., 2020; Choi et al., 2018], and the validity of this assumption is empirically supported by their numerical evidence. On the other hand, this assumption is indeed debatable. As we mentioned in our "limitations" section, when one sample in the X-domain corresponds to many samples in the Y-domain, this assumption does not hold. In such cases, different approaches should be considered. There do exist weakened versions in the literature, including those having probabilistic characteristics [Huang et al., 2018; Choi et al., 2020, Yang et al., 2023], but they require different treatments and thus are out of scope. As the invertibility assumption does have many success examples in the literature, we consider it worth studying and understanding. Nonetheless, it should not be considered a "universal" model for UDT problem, due to its limitations. We have added the references [Huang et al., 2018; Choi et al., 2020, Yang et al., 2023] in the "Limitations" section in case that some readers are interested.
>
>
> (To be continued...)

---

> ### Author Response · Authors · 2023-11-17
>
> ### ***Response to Reviewer siTh (Part 2)***
>
> &nbsp;
>
> **Q2.** The reviewer has made a great point. We should mention that Proposition 1 presents a sufficient condition under which MPA exist for the given data distribution. However, this condition is by no means necessary. In fact, the existence of MPA can be proved under different sufficient conditions, e.g., Proposition 2.6 in [Moriakov et al, 2020].
>
> When ${\mathcal{X}}$ may not be simply connected, we can reformulate Proposition 1 to work under more relaxed condition. The following modified proposition incorporates such a case.
>
> **Proposition:**
>     Suppose that $P_x$ admits a continuous PDF, $p({\boldsymbol{x}})$, and $p({\boldsymbol{x}})>0, \forall {\boldsymbol{x}} \in {\mathcal{X}}$. Assume that there exists a simply connected subset of ${\mathcal{X}}$ with strictly positive measure. Then, there exists a continuous non-trivial (non-identity) ${\boldsymbol{h}}(\cdot)$ such that ${\boldsymbol{h}}* P_x = P_x$. (here ${\boldsymbol{h}}* P_x$ denotes the push-forward of measure $P_x$ by the function ${\boldsymbol{h}}$)
>
> We give a concise proof here:
>
> **Proof:**
> The proof involves transforming the PDF on the simply connected subset to a uniform distribution, followed by using MPA of uniform distribution, and finally inverting the uniform distribution to the original PDF.
>
> Let ${\mathcal{A}} \subseteq {\mathcal{X}}$ denote the largest simply connected subset of ${\mathcal{X}}$ with strictly positive measure. We construct ${\boldsymbol{m}}: {\mathcal{A}} \to {\mathcal{A}}$ by reducing the problem of finding MPA on ${\mathcal{A}}$ to finding an MPA on the uniform distribution. To that end, we first define a proxy distribution $\overline{P}_{{\boldsymbol{x}}}$ supported on ${\mathcal{A}}$, such that:
> \begin{align}  \overline{P}_x[{\mathcal{B}}] = \begin{cases}\frac{1}{P_x [{\mathcal{A}}]} P_x[{\mathcal{B}}], \quad \forall {\mathcal{B}} \subseteq {\mathcal{A}}  \\\
> 0, \quad \text{ otherwise.}
> \end{cases}\end{align}
>
> Then, using the same approach as in the proof of Lemma 1 in the manuscript, using Darmois construction, one can construct a function ${\boldsymbol{d}}: {\mathcal{A}} \to (-1,1)^{D_x}$ that maps $\overline{P}_{{\boldsymbol{x}}}$ to a uniform distribution. Next, one can form a continuous mapping ${\boldsymbol{m}}: {\mathcal{A}} \to {\mathcal{A}}$ as follows:
> $$ {\boldsymbol{m}} = {\boldsymbol{d}}^{-1} \circ {\boldsymbol{m}}_U \circ {\boldsymbol{d}},$$
> where ${\boldsymbol{m}}_U:(-1,1)^{D_x} \to (-1,1)^{D_x} $ is a continuous MPA on the uniform distribution over $(-1,1)^{D_x}$.
>
> In order to construct MPA ${\boldsymbol{h}}: {\mathcal{X}} \to {\mathcal{X}}$ using ${\boldsymbol{m}}: {\mathcal{A}} \to {\mathcal{A}}$, we define ${\boldsymbol{h}}$ as follows:
>
> \begin{align} \boldsymbol{h} ({\boldsymbol{x}})= \begin{cases} \boldsymbol{m}({\boldsymbol{x}}), \quad \text{if } \boldsymbol{x} \in {\mathcal{A}} \\\
> {\boldsymbol{x}}, \quad \text{otherwise.}
> \end{cases}\end{align}
>
> Now, for ${\boldsymbol{h}}$ to be continuous, we need to ensure that ${\boldsymbol{m}}({\boldsymbol{x}}) = {\boldsymbol{x}}, \forall {\boldsymbol{x}} \in {\rm bd}({\mathcal{A}})$, where ${\rm bd}({\mathcal{A}})$ is the boundary of ${\mathcal{A}}$. This can be done by constructing ${\boldsymbol{m}}_U$ following the procedure described in [Hyvärinen and Pajunen, 1999, Section 2.2].
>
> &nbsp;
>
> ### **References**
>
> [Choi et al., 2018] Choi, Y., Choi, M., Kim, M., Ha, J.-W., Kim, S., and Choo, J. (2018). StarGAN: Unified
> generative adversarial networks for multi-domain image-to-image translation. In Proceedings of IEEE/CVF
> Computer Vision and Pattern Recognition (CVPR), pages 8789–8797.
>
> [Choi et al., 2020] Choi, Y., Uh, Y., Yoo, J., and Ha, J.-W. (2020). StarGAN v2: Diverse image synthesis for
> multiple domains. In Proceedings of IEEE/CVF Computer Vision and Pattern Recognition (CVPR), pages
> 8188–8197.
>
> [Gabbay et al., 2021] Gabbay, A., Cohen, N., and Hoshen, Y. (2021). An image is worth more than a thousand
> words: Towards disentanglement in the wild. Advances in Neural Information Processing Systems, 34:9216–
> 9228.
>
> [Gabbay and Hoshen, 2021] Gabbay, A. and Hoshen, Y. (2021). Scaling-up disentanglement for image translation.
> In Proceedings of the IEEE/CVF International Conference on Computer Vision, pages 6783–6792.
>
> [Huang et al., 2018] Huang, X., Liu, M.-Y., Belongie, S., and Kautz, J. (2018). Multimodal unsupervised
> image-to-image translation. In Proceedings of European Conference on Computer Vision (ECCV), pages
> 172–189.
>
> [Hyvärinen and Pajunen, 1999] Hyv¨arinen, A. and Pajunen, P. (1999). Nonlinear independent component
> analysis: Existence and uniqueness results. Neural networks, 12(3):429–439.
>
> [Jung et al., 2022] Jung, C., Kwon, G., and Ye, J. C. (2022). Exploring patch-wise semantic relation for
> contrastive learning in image-to-image translation tasks. In Proceedings of the IEEE/CVF Conference on
> Computer Vision and Pattern Recognition (CVPR), pages 18260–18269.
>
> (To be continued...)

---

> ### Author Response · Authors · 2023-11-17
>
> ## ***Response to Reviewer siTh (Part 3)***
>
> &nbsp;
>
> ### **References**
>
>
> [Kim et al., 2020] Kim, J., Kim, M., Kang, H., and Lee, K. (2020). U-GAT-IT: Unsupervised generative
> attentional networks with adaptive layer-instance normalization for image-to-image translation. In Proceedings
> of International Conference on Learning Representations (ICLR).
>
> [Ko et al., 2023] Ko, K., Yeom, T., and Lee, M. (2023). Superstargan: Generative adversarial networks for
> image-to-image translation in large-scale domains. Neural Networks, 162:330–339.
>
> [Liu et al., 2017] Liu, M.-Y., Breuel, T., and Kautz, J. (2017). Unsupervised image-to-image translation networks.
> In Advances in Neural Information Processing Systems (NeurIPS), volume 30.
>
> [Moriakov et al., 2020] Moriakov, N., Adler, J., and Teuwen, J. (2020). Kernel of CycleGAN as a principle
> homogeneous space. In Proceedings of International Conference on Learning Representations (ICLR).
>
> [Park et al., 2023] Park, J., Kim, S., Kim, S., Cho, S., Yoo, J., Uh, Y., and Kim, S. (2023). Lanit: Language-
> driven image-to-image translation for unlabeled data. In Proceedings of the IEEE/CVF Conference on
> Computer Vision and Pattern Recognition, pages 23401–23411.
>
> [Park et al., 2020] Park, T., Efros, A. A., Zhang, R., and Zhu, J.-Y. (2020). Contrastive learning for unpaired
> image-to-image translation. In Proceedings of European Conference on Computer Vision (ECCV), pages
> 319–345.
>
> [Yang et al., 2023] Yang, S., Jiang, L., Liu, Z., and Loy, C. C. (2023). Gp-unit: Generative prior for versatile
> unsupervised image-to-image translation. IEEE Transactions on Pattern Analysis and Machine Intelligence.
>
> [Zhu et al., 2017] Zhu, J.-Y., Park, T., Isola, P., and Efros, A. A. (2017). Unpaired image-to-image translation
> using cycle-consistent adversarial networks. In Proceedings of IEEE/CVF Computer Vision and Pattern
> Recognition (CVPR), pages 2223–2232.

---

> > ### Comment · Reviewer_siTh · 2023-11-17
> > **Reviewer Response**
> >
> > Thanks a lot for the explanations. I have read the authors responses and the other reviews. Regarding the choice of baselines I have found the authors response compelling and I decided to raise the confidence in the score.

---

### Official Review · Reviewer_bENZ · 2023-11-01

**Soundness:** 3 good
**Presentation:** 3 good
**Contribution:** 3 good
**Rating:** 6
**Confidence:** 4

**Summary:**

This paper seeks to address the issue of content misalignment in unsupervised domain translation. The authors pinpoint the presence of "measure preserving automorphism" (MPA) as the primary culprit and present a theoretically-founded method to neutralize it. Their method was validated on datasets like Edges to Rotated Shoes, yielding high-quality samples that maintained content integrity.

**Strengths:**

1. It is innovative to introduce auxiliary variables for tackling the MPA issue. I'd like to offer more insights on this approach. Essentially, **supervised domain translation can be seen as a specific instance of their method.** By choosing a specific auxiliary variable, we can tailor each conditional distribution $p(x|u=u_i)$ to hold precisely one sample, $x_i$, with a probability of 1, whereas all other samples in the space have a zero probability. Similarly, we manipulate each conditional distribution $p(y|u=u_i)$ to include only one sample, $y_i$, also with a probability of 1. These corresponding sample pairs, $(x_i, y_i)$, are essentially the supervised pairs for domain translation. By adjusting loss function 7 (i.e., the distance metric of cycle loss and the balance parameter $\lambda$), this approach could replicate any supervised domain translation methods. The paper's impact would be significantly enhanced if the authors included this observation.

2. Overall, the structure of the proof is clear and easy to follow.
3. The experiments verified that this method could generate content-preserved samples with high quality, which corroborates their theory.

**Weaknesses:**

Overall, the structure of the theory is clear. However, there are several mistakes that should be corrected:
1. The MPA of the PDF of a gaussian distribution $N(\mu, \sigma)$ should be $h(x) = 2\mu - x$, rather than $h(x) = \mu - x$.
2. Within the "Notation" section of the introduction, "A" ought to be a subset of "Y", not "X".

Honestly, it is impractical to check every detail of the proof. The author should ensure the proof's rigor and review it meticulously.

Additional suggestion: Assumption 1 is confusing. I think it refers to the existence of the content-preserved mapping $f^*$ and $y^*$. Please make it clearer.

**Questions:**

In this study, it appears that only one auxiliary variable is used to diversify the distribution. What would happen if we used several variables? For example, we're considering not just the distributions $p(x|u=u_i)$ and $p(y|u=u_i)$, but also $p(x|v=v_j)$ and $p(y|v=v_j)$. Intuitively, utilizing one auxiliary variable is akin to "slicing" the original distribution in one way, while employing multiple variables is like trying different ways to make the "cuts".

---

> ### Author Response · Authors · 2023-11-17
>
> ## ***Response to Reviewer bENZ***
>
>
> &nbsp;
>
> ### **[Response to Strengths]**
>
> Thank you for the appreciation for our work. Regarding the relation to supervised domain translation, the reviewer made an interesting point. Matching distributions between $P_{x|u=u_i}$ and ${\boldsymbol{f}}{ * P_{y|u=u_i}}$ (here  ${\boldsymbol{f}}{ * P_{y|u=u_i}}$ denote the push forward of measure  $ P_{y|u=u_i}$ by the function  ${\boldsymbol{f}}$), where both $P_{x|u=u_i}$ and $P_{y|u=u_i}$ are Dirac delta distributions (as they are defined over a single sample), will be equivalent to enforcing ${\boldsymbol{x}}^{(i)} = {\boldsymbol{f}} ({\boldsymbol{y}}^{(i)})$, where ${\boldsymbol{x}}^{(i)} \sim P_{x|u=u_i}$ and ${\boldsymbol{y}}^{(i)} \sim P_{y|u=u_i}$. Therefore, the objective will be equivalent to minimizing $ \sum_{i=1}^N || {\boldsymbol{x}}^{(i)} - {\boldsymbol{f}}({\boldsymbol{y}}^{(i)}) ||_2^2.$ This indeed makes the distribution matching problem boil down to a sample matching problem. This observation is indeed nice. We have run out of space in the main paper. But we have added a section "Additional Remark" in the supplementary material to share this observation.
>
>
> &nbsp;
> ### **Weaknesses**
>
> **1.** Thank you for your careful reading, and catching the mistake. Indeed, it should be $h(x) = 2 \mu - x$, instead of $h(x) = \mu - x$. We have made this correction in the revised version.
>
>
> **2.** Thank you for your careful reading. We have corrected this mistake in the revised version.
>
>
> **[Proof Review]**
>
> We appreciate the suggestion. We have cross checked the proofs for several times among the authors. Once the review process concludes, we will again thoroughly read through the proofs as well as the main paper to further ensure the proof's mathematical rigor.
>
> **[Unclear Assumption 1]**
>
> We agree. We have re-written this part and removed some repetitions. Now Assumption 1 starts with assuming the existence of ${\boldsymbol{f}}^\star,{\boldsymbol{g}}^\star$, following the reviewer's suggestion.
>
> &nbsp;
>
> ### **Questions**
>
> **[Multiple Auxiliary Variables]**
>
> The reviewer has made an interesting and valid point.
> Indeed, one can use multiple auxiliary variables in order to achieve greater diversity of the conditional distributions. In principle, multiple auxiliary variables can be regarded as a single joint auxiliary variable. For example, if $v$ and $u$ are two auxiliary variables, one can introduce a new auxiliary variable $w$ whose alphabet is the Cartesian product of the alphabets of $v$ and $u$, i.e., $w  \in \\{ v_1,\ldots,v_I \\}\times \\{u_1,\ldots,u_J\\}$. Therefore, the theoretical analysis remains unchanged---but the increased size of the alphabet does make the identifiability holds with a higher probability, as in presented in Theorems 1 and 2. In the experiments, we have actually used multiple auxiliary variables following the above way; see the experiments on CelebA-HQ vs. Bitmoji. There, black/non-black hair color corresponds to one auxiliary variable, whereas gender corresponds to another auxiliary variable.

---

> > ### Comment · Reviewer_bENZ · 2023-11-22
> > **Thanks for your comments**
> >
> > Thanks for your comments. I will keep my score as 6.

---

### Official Review · Reviewer_4LhS · 2023-11-07

**Soundness:** 3 good
**Presentation:** 3 good
**Contribution:** 2 fair
**Rating:** 5
**Confidence:** 4

**Summary:**

The paper aims to propose an unsupervised image translation framework that ensures identifiability of underlying generator maps. It achieves the same by relying on auxiliary variables. Promising empirical results showcase the potential of the method on benchmark image datasets.

**Strengths:**

The writing of the paper is good with detailed exposition of the problem. It also includes detailed notes on related literature. The paper produces promising qualitative and quantitative results based on experiments. It also gives ample ablation and suggestions on the architecture and parameters involved. The brief declaration of limitations is appreciated.

**Weaknesses:**

The theory under quite strong assumptions tends to be straightforward and does not fully complement what the paper set out to achieve. It revolves around a particular model and due to certain vague notions becomes somewhat vacuous. The empirical results give the paper strength which the theory fails to support. In my opinion, the experiments should be prioritized.

**Questions:**

1. My first concern is regarding the strong assumption that continuous functions $f^*$  and  $g^*$ exist under arbitrary input data on both domains. It is quite challenging to ensure that the optimal transport map between distributions has any regularity (e.g. Lipschitz continuity). Also, in most cases proving so requires the support of the base distribution to have restrictions in terms of convexity. In real data, the same hardly follows. Perhaps sacrificing generality for the sake of accuracy would be better for the theory.

2. The discussion on the notion of "content" seems vague. Is the content of an image and its rotated counterpart the same? Is there any generalization of it for general group actions?

Is there any way of justifying Assumption 1, even if with examples?

It seems to me that homeomorphic spaces will have the same "content", whereas the assumption claims the converse. Am I right in saying that?

3. Given there exist non-unique members in the kernel (i.e. multiple solutions bringing about zero loss), is Definition 1 even meaningful in a non-parametric setup, where there is no inherent identifier?

4. The entire theory revolves around the CycleGAN loss in particular. This does not complement the initial impression of unsupervised domain translation in general. The proposed loss function ($7$) is also a modified CycleGAN setup. Also, what are the "any criterion" in Fact 1?

Does the discriminator play any role in ensuring identifiability? This seems crucial as the resultant translation map would be a result of a stable discriminator.

5. [Section 3] Can this at all be called unsupervised given that pseudo or weak labels ($u$) are used? Also, what is meant by "sufficiently different" $P_{x|u_i}$ and $P_{x|u_j}$?

Shouldn't the difference between distributions $P_{x|u(A,B)}[A]$ and $P_{x|u(A,B)}[B]$ be based on a divergence measure and not inequality ($\neq$)?

There remain some typographical/grammatical errors in the manuscript (e.g. see the Section Identifiability Characterization).

---

> ### Author Response · Authors · 2023-11-17
>
> ## ***Response to Reviewer 4LhS (Part 1/5)***
> &nbsp;
>
> ## Overall Response:
>
> There are some perspectives that we do share with the reviewer, e.g., regarding the scope of the model. However, there are some points that we respectfully disagree with, e.g., the importance of our theoretical results - and we hope our clarification could narrow the gap.
> Nonetheless, we appreciate the reviewer's feedback, which does reflect careful reading, sharp assessment and constructive criticism, despite some different opinions. The comments also pushed us to reconstruct the assumption part for clarity, which we hope could help alleviate concerns or confusion.
>
> &nbsp;
> ## **Questions**:
>
> &nbsp;
> ### **Q1.**
> **[Optimal Transport]**
>
> We first hope to clarify that our assumption and learning objective are not related to optimal transport. We take responsibility of not making the assumption clear enough. To clarify, Assumption 1 assumes that there exist "ground-truth" invertible mappings ${\boldsymbol{f}}^\star$ and ${\boldsymbol{g}}^\star$ that link data pairs $({\boldsymbol{x}},{\boldsymbol{y}})$ who represent the same entity (e.g., the digit "3"). The mappings need not to be the optimal transport mappings. For example, between the MNIST digits and rotated MNIST digits, it is unlikely that the rotation is an optimal transport map.  Note that recovering the ground-truth ${\boldsymbol{f}}^\star$ and ${\boldsymbol{g}}^\star$ also means finding the right ${\boldsymbol{x}},{\boldsymbol{y}}$ correspondence. In our example, where data from ${\mathcal{X}}$ and ${\mathcal{Y}}$ are order-shuffled digits, this does not necessarily mean using the functions to move the least amount of mass of the data distribution in each domain. In this example, the optimal transport map could be one of the MPAs that we do not hope to find.
>
> In the revised version, we have rewritten and reconstructed the part around Assumption 1. We hope this could make the assumption clearer.
>
>
> **[Continuity of the Function in Assumption 1]**
>
> The existence of Lipschitz continuous mappings to attain good translation performance may be supported by empirical results observed in CycleGAN and variants -- i.e. the fact that CycleGAN and variants can often find deterministic and weight-bounded neural networks to attain good translation performance. Note that a neural network can be viewed as a continuous function. If the weights are bounded, then the function is Lipschitz continuous [Bartlett et al., 2017].
>
> **[Why interested in Model under Assumption 1]**
>
> Although we do not consider optimal transports, we actually do not object to the comment that Assumption 1 is relatively stringent, as the existence of one-to-one mapping between the two domains is debatable---which we also discussed in "limitations". Our interest in understanding the $({\boldsymbol{f}}^\star,{\boldsymbol{g}}^\star)$-identfiability properties of the UDT model under Assumption 1 is driven by the following reasons: First, the model is an empirically very successful one in the domain of computer vision. Many UDT works, especially  CycleGAN [Zhu et al., 2017] and its variants [Liu et al., 2017; Choi et al., 2018, Park et al., 2020; Kim et al., 2017], essentially use the model in Assumption 1. These works seek continuous deterministic functions that translate between the two domains. These works have enjoyed impressive empirical success in practice, which suggests that this model is a **useful model**. Hence, although the model has several debatable assumptions, the usefulness of the model in practice still makes it interesting and worth studying. Second, even under the relatively stringent conditions as in Assumption 1, it had been unclear what are the key factors that could make CycleGAN-like methods fail or succeed. Although there have been many attempts to understand the identifiability of this model [Galanti et al., 2018b, Galanti et al., 2018a, Moriakov et al, 2020], no solution has been put forward to underpin the identifiability of ${\boldsymbol{f}}^\star$ and ${\boldsymbol{g}}^\star$. Hence, our intention in this work is not to question the model, despite the awareness of its potential limitations. Our intention is to offer understanding assuming that the model holds.
> We hope the reviewer agrees that this intention is meaningful given that the model is so widely used. Of course, we agree with reviewer that we should articulate the reason why the model itself may be a stringent one - but providing a more general model is a different topic.
>
> We have updated the manuscript according to the above discussion. In particular, we followed the reviewer's comment to articulate the reason why some assumptions may be stringent, and added a comment on Lipchitz continuity assumption in Theorem 2. In addition, we explained the reason why understanding this model is still of interest.
>
> (To be continued...)

---

> ### Author Response · Authors · 2023-11-17
>
> ## ***Response to Reviewer 4LhS (Part 2/5)***
> &nbsp;
>
> ### **Q2.**
>
> **["Content"]**
>
> The reviewer has made a good point. We agree with the reviewer that the use of the terminology "content" should have been more cautious as the term *per se* does not have any mathematical meaning, which may be a source of confusion.
>
> To alleviate the confusion, we have done some reconstruction of the part around Assumption 1 (and also rewrote Assumption 1 in a more careful way).
> To clarify, our definition of "content" is based on the unique correspondence: We consider a setting where
> for every ${\boldsymbol{x}}\in {\mathcal{X}}$, there exists exact one corresponding ${\boldsymbol{y}}\in{\mathcal{Y}}$, and vice versa. In addition, there exist deterministic continuous functions ${\boldsymbol{f}}^{\star}: {\mathcal{Y}} \rightarrow {\mathcal{X}}$ and ${\boldsymbol{g}}^\star: {\mathcal{X}} \rightarrow {\mathcal{Y}}$ functions that link the pairs; i.e.,
> \begin{align}
> {\boldsymbol{f}}^\star({\boldsymbol{y}}) ={\boldsymbol{x}},\quad {\boldsymbol{g}}^\star({\boldsymbol{x}}) = {\boldsymbol{y}},~~\forall~\text{pair $({\boldsymbol{x}},{\boldsymbol{y}})$}.
> \end{align}
>
> In the context of domain translation, a linked $({\boldsymbol{x}},{\boldsymbol{y}})$ pair can be regarded as cross-domain data samples that represent the same "content", and the translation functions $({\boldsymbol{f}}^\star,{\boldsymbol{g}}^\star)$ are responsible for changing their "appearances/styles".
> Under our settings, the two cross-domain data samples share the same content means that ${\boldsymbol{f}}^\star({\boldsymbol{x}})={\boldsymbol{y}}$ and ${\boldsymbol{g}}^\star({\boldsymbol{y}})={\boldsymbol{x}}$, i.e., they represent the same entity and are linked through the ground-truth ${\boldsymbol{f}}^\star$ and ${\boldsymbol{g}}^\star$. In this sense, "content" is the same as "entity". Hence, the content of an image and its rotated version are the same, under this definition.
> We have incorporated the above discussion in the new version.
>
> **[Examples of Assumption 1]**
>
> Assumption 1 assumes the existence of continuous "ground-truth" translation functions that can map between the correspondence pairs ${\boldsymbol{x}}, {\boldsymbol{y}}$. For example, for the task of writing style translation, a function that can translate a sentence written in one style to the same sentence in another style and vice versa are ${\boldsymbol{f}}^\star$ and ${\boldsymbol{g}}^\star$. In MNIST digits to rotated MNIST digits example, ${\boldsymbol{f}}^\star$ and ${\boldsymbol{g}}^\star$ correspond to positive and negative 90 degrees rotations.
>
> **[homeomorphic spaces will have the same "content"?]**
>
> In the context of previous paragraphs regarding Assumption 1 and content, one can see that
> content is not a property of the data spaces ${\mathcal{X}}$ and ${\mathcal{Y}}$.
> Therefore, it would be more accurate to say whether specific sample pair $({\boldsymbol{x}}, {\boldsymbol{y}})$ share the same content, instead of whether the spaces ${\mathcal{X}}$ and ${\mathcal{Y}}$ share the same content.
>
> &nbsp;
>
> ### **Q3.**
>
> We believe this is a key point that is really worth clarifying. Note that the problem of $({\boldsymbol{f}}^\star,{\boldsymbol{g}}^\star)$-identifiability is only associated with the ground-truth data model, assuming that there exists a pair of ground-truth $({\boldsymbol{f}}^\star,{\boldsymbol{g}}^\star)$ (in our case, Assumption 1). The fact that the kernel has multiple solutions is the challenge that we exactly aim to deal with. The key of this setup that there exist a pair of ground-truth ${\boldsymbol{f}}^\star,{\boldsymbol{g}}^\star$, which can be understood as the latent parameters that we hope to estimate. Our research question is "how to find the ground-truth data ${\boldsymbol{f}}^\star,{\boldsymbol{g}}^\star$ in the underlying data model among all possible solutions in the kernel?" -- the notion of "identifiability" means that we hope to estimate (identify) the exact ground-truth ${\boldsymbol{f}}^\star,{\boldsymbol{g}}^\star$. In other words, the CycleGAN objective in (3) has multiple solutions that bring about zero loss. This means that the solution returned by solving (3) may or may not correspond to ${\boldsymbol{f}}^\star$ and ${\boldsymbol{g}}^\star$. That is, optimizing the vanilla CycleGAN objective does not ensure the identification of ${\boldsymbol{f}}^\star$ and ${\boldsymbol{g}}^\star$. This is the core non-identifiability issue that we address in this work.
>
> In a nutshell, Definition 1 specifies our (identifiability) goal. Our sense is that it is meaningful under the setting of Assumption 1 (i.e., assuming that there are ground-truth $({\boldsymbol{f}}^\star,{\boldsymbol{g}}^\star)$). The definition helps understand the reason why CycleGAN objective (3) does not ensure recovery of the ground-truth ${\boldsymbol{f}}^\star$ and ${\boldsymbol{g}}^\star$. It also helps understand why MPA is harmful in the context of translation.
>
> (To be continued....)

---

> ### Author Response · Authors · 2023-11-17
>
> ## ***Response to Reviewer 4LhS (Part 3/5)***
>
> &nbsp;
>
> ### **Q4.**
>
> **[Why We are Interested in GAN-Based Distribution Matching]**
>
> The reviewer is correct that our development is very much driven by the empirical successes of CycleGAN and its variants. However, we argue that our theoretical understanding is of broader interest beyond CycleGAN. Let us explain.
> First, understanding the key module, i.e., GAN-based distribution matching, in CycleGAN and variants, is actually essential for unpaired domain translation, as a vast volume of UDT methods are based on distribution matching. It is worth noting that our proof applies to any distribution criteria, while GAN is one of the *realizations* of distribution matching. Second, distribution matching modules are not only used in UDT. It is also the key module for tasks such as domain adaptation [Ben-David et al., 2010, Ganin et al., 2016, Gulrajani and Hashimoto, 2022] and transfer learning [Long et al., 2013; Zhuang et al., 2020]. Our understanding to this module can be easily used for those applications.
>
>
> **[Clarification on "Any Criterion"]**
>
> The proof of Fact 1 relies on the estimates of ${\boldsymbol{f}}^\star$ and ${\boldsymbol{g}}^\star$ (i.e., $\widehat{{\boldsymbol{f}}}$ and $\widehat{{\boldsymbol{g}}}$, respectively) satisfying  $P_x = \widehat{\boldsymbol{f}}* P_y $ and $ P_y = \widehat{{\boldsymbol{g}}} * P_x $ ( the notation $\widehat{\boldsymbol{f}}* P_y$ denotes the push-forward of measure $P_y$ by function $ \widehat{\boldsymbol{f}} $ ), and $ \widehat{\boldsymbol{f}} $ and $ \widehat{\boldsymbol{g}} $ being invertible functions. This can be achieved by many objective functions other than (3). For example, distribution matching could be achieved by other methods other than using GAN, such as moment matching. Also, invertibility can be achieved by using invertible neural networks, entropy maximization, contrastive learning, just to name a few. Hence, "any criterion" refers to all possible learning criteria for realizing the distribution matching and invertibility enforcing.
>
>
> **[Discriminator's Role in Identifiability]**
>
> The discriminator is an indispensable part for realizing GAN-based distribution matching, but the specific network architecture was not taken into consideration in the analysis. We assume that the optimal solution to (7) is obtained in our identifiability analysis and that all the involved neural networks are universal representation learners. As a result, perfect distribution can be matched by solving the GAN objective, as proven in [Goodfellow et al., 2014]. Hence, to have the results in Lemma 1, the discriminator should be part of an optimal solution of the minimax problem (7). As we aim to characterize how the learning criterion works at its best, it is reasonable to analyze under this setting. Note that analyzing optimal solutions of the learning criteria is common practice in the identifiability-related research  [Von Kugelgen et al., 2022; Lyu et al., 2022; Wang and Isola, 2020].
>
> &nbsp;
>
> ### **Q5.**
>
> **[Terminology of "Unsupervised"]**
>
> We completely agree with the reviewer that the term is debatable. We are aware of the possible confusion, but we chose to use "unsupervised" because in domain translation, "supervision" normally mean cross domain data pair alignment [Isola et al., 2017]
> "Semi-supervised" or "Weak supervised" DT may be confused with those methods who use partial data alignment information [Wang et al., 2020; Mustafa and Mantiuk, 2020]. Our method does use some additional auxiliary information, but it still does not require data pair alignment information. This is the reason why we chose to stick with "UDT". That being said, we agree that the term can cause confusion. In the revised version, we have added a footnote to reflect the above discussion - i.e., we emphasize that "unsupervision" means that no data alignment is needed, but we acknowledge that we do use more information than classical frameworks such as CycleGAN.
>
>
> (To be continued...)

---

> ### Author Response · Authors · 2023-11-17
>
> ## ***Response to Reviewer 4LhS (Part 4/5)***
>
> &nbsp;
>
> **[The Meaning of "Sufficiently Different"]**
>
> The exact definition for sufficiently diverse distributions is in Definition 4. As we mentioned in the manuscript, generally speaking, sufficiently diverse condition on a set of distribution requires that their PDFs exhibit different shapes in their support. Note that divergence measures, such as KL-divergence and Wasserstein distance, cannot characterize the sufficiently diverse conditions that we state in Definitions 4 and 5. For example, two uniform distributions ${\mathcal{U}}([-1, 1])$ and ${\mathcal{U}}([-2, -2])$ has large KL-divergence and Wasserstein distance. However, they share a common MPA, reflection about the origin, and therefore are not sufficiently diverse in the sense of Definition 4. We hope to mention that similar assumptions on distribution diversity (either in the form of probability mass assigned to measurable sets or gradients of PDF) are not uncommon in the machine learning literature [Zheng et al., 2022; Hyvarinen et al., 2019; Khemakhem et al., 2020; Zheng and Zhang, 2023].
>
>
>
> **[Typographical/Grammatical Errors]**
>
> We thank the reviewer for their careful reading and spotting errors. We have proofread and corrected some typographical errors following the reviewer's comment. We will do more rounds of proofreading to correct the remaining errors before we submit the final version.
>
>
> &nbsp;
>
> ### **Summary**
>
> We thank the reviewer for giving us the feedback that pushed us to make the model assumptions and our goal clearer. To re-iterate, our paper can be understood as a pipeline of "model hypothesis -- theory development based on the model -- empirical validation of the theory". The model hypothesis part (Assumption 1) was a summary of and supported by empirically successful works such as CycleGAN and its variants. Our goal is to develop a theory that offers understanding to UDT methods under the model - why they sometimes fail and how to fix these pathological cases? We do agree that the model can be restrictive in some cases, but the model is useful and thus understanding it is a meaningful task. We hope the clarifications could help alleviate the reviewer's concerns, and that the reviewer could reconsider the score.
>
> &nbsp;
>
> ### **References**
>
> [Bartlett et al., 2017] Bartlett, P. L., Foster, D. J., and Telgarsky, M. J. (2017). Spectrally-normalized margin
> bounds for neural networks. Advances in neural information processing systems, 30.
>
> [Ben-David et al., 2010] Ben-David, S., Blitzer, J., Crammer, K., Kulesza, A., Pereira, F., and Vaughan, J. W.
> (2010). A theory of learning from different domains. Machine learning, 79:151–175.
>
> [Choi et al., 2018] Choi, Y., Choi, M., Kim, M., Ha, J.-W., Kim, S., and Choo, J. (2018). StarGAN: Unified
> generative adversarial networks for multi-domain image-to-image translation. In Proceedings of IEEE/CVF
> Computer Vision and Pattern Recognition (CVPR), pages 8789–8797.
>
> [Galanti et al., 2018a] Galanti, T., Benaim, S., and Wolf, L. (2018a). Generalization bounds for unsupervised
> cross-domain mapping with WGANs. arXiv preprint arXiv:1807.08501.
>
> [Galanti et al., 2018b] Galanti, T., Wolf, L., and Benaim, S. (2018b). The role of minimal complexity functions
> in unsupervised learning of semantic mappings. In Proceedings of International Conference on Learning
> Representations (ICLR).
>
> [Ganin et al., 2016] Ganin, Y., Ustinova, E., Ajakan, H., Germain, P., Larochelle, H., Laviolette, F., Marchand,
> M., and Lempitsky, V. (2016). Domain-adversarial training of neural networks. Journal of Machine Learning
> Research (JMLR), 17:2096–2030.
>
> [Goodfellow et al., 2014] Goodfellow, I., Pouget-Abadie, J., Mirza, M., Xu, B., Warde-Farley, D., Ozair, S.,
> Courville, A., and Bengio, Y. (2014). Generative adversarial networks. In Advances in Neural Information
> Processing Systems (NeurIPS).
>
> [Gulrajani and Hashimoto, 2022] Gulrajani, I. and Hashimoto, T. (2022). Identifiability conditions for domain
> adaptation. In Proceedings of International Conference on Machine Learning (ICML), pages 7982–7997.
>
> [Hyvarinen et al., 2019] Hyvarinen, A., Sasaki, H., and Turner, R. (2019). Nonlinear ica using auxiliary variables
> and generalized contrastive learning. In Proceedings of International Conference on Artificial Intelligence and
> Statistics (AISTATS), pages 859–868. PMLR.
>
> [Isola et al., 2017] Isola, P., Zhu, J.-Y., Zhou, T., and Efros, A. A. (2017). Image-to-image translation with
> conditional adversarial networks. In Proceedings of IEEE/CVF Computer Vision and Pattern Recognition
> (CVPR), pages 1125–1134.
>
> [Khemakhem et al., 2020] Khemakhem, I., Kingma, D., Monti, R., and Hyvarinen, A. (2020). Variational
> autoencoders and nonlinear ICA: A unifying framework. In International Conference on Artificial Intelligence
> and Statistics, pages 2207–2217. PMLR.

---

> ### Author Response · Authors · 2023-11-17
>
> ## ***Response to Reviewer 4LhS (Part 5/5)***
>
> &nbsp;
>
> ### **References**
>
> [Kim et al., 2017] Kim, T., Cha, M., Kim, H., Lee, J. K., and Kim, J. (2017). Learning to discover cross-domain
> relations with generative adversarial networks. In Proceedings of International Conference on Machine Learning
> (ICML), pages 1857–1865.
>
> [Liu et al., 2017] Liu, M.-Y., Breuel, T., and Kautz, J. (2017). Unsupervised image-to-image translation networks.
> In Advances in Neural Information Processing Systems (NeurIPS), volume 30.
>
> [Long et al., 2013] Long, M., Wang, J., Ding, G., Sun, J., and Yu, P. S. (2013). Transfer feature learning with
> joint distribution adaptation. In Proceedings of the IEEE international conference on computer vision, pages
> 2200–2207.
>
> [Lyu et al., 2022] Lyu, Q., Fu, X., Wang, W., and Lu, S. (2022). Understanding latent correlation-based
> multiview learning and self-supervision: An identifiability perspective. In Proceedings of International
> Conference on Learning Representations (ICLR).
>
> [Moriakov et al., 2020] Moriakov, N., Adler, J., and Teuwen, J. (2020). Kernel of CycleGAN as a principle
> homogeneous space. In Proceedings of International Conference on Learning Representations (ICLR).
>
> [Mustafa and Mantiuk, 2020] Mustafa, A. and Mantiuk, R. K. (2020). Transformation consistency regularization:
> A semi-supervised paradigm for image-to-image translation. In Proceedings of the IEEE/CVF Conference on
> Computer Vision and Pattern Recognition (CVPR), pages 599–615.
>
> [Park et al., 2020] Park, T., Efros, A. A., Zhang, R., and Zhu, J.-Y. (2020). Contrastive learning for unpaired
> image-to-image translation. In Proceedings of European Conference on Computer Vision (ECCV), pages
> 319–345.
>
> [Von K¨ugelgen et al., 2021] Von K¨ugelgen, J., Sharma, Y., Gresele, L., Brendel, W., Sch¨olkopf, B., Besserve,
> M., and Locatello, F. (2021). Self-supervised learning with data augmentations provably isolates content from
> style. In Advances in Neural Information Processing Systems (NeurIPS), volume 34, pages 16451–16467.
>
> [Wang and Isola, 2020] Wang, T. and Isola, P. (2020). Understanding contrastive representation learning through
> alignment and uniformity on the hypersphere. In International Conference on Machine Learning, pages
> 9929–9939. PMLR.
>
> [Wang et al., 2020] Wang, Y., Khan, S., Gonzalez-Garcia, A., Weijer, J. v. d., and Khan, F. S. (2020). Semi-
> supervised learning for few-shot image-to-image translation. In Proceedings of the IEEE/CVF Conference on
> Computer Vision and Pattern Recognition (CVPR), pages 4453–4462.
>
> [Zheng et al., 2022] Zheng, Y., Ng, I., and Zhang, K. (2022). On the identifiability of nonlinear ica: Sparsity
> and beyond. Advances in Neural Information Processing Systems, 35:16411–16422.
>
> [Zheng and Zhang, 2023] Zheng, Y. and Zhang, K. (2023). Generalizing nonlinear ica beyond structural sparsity.
> arXiv preprint arXiv:2311.00866.
>
> [Zhu et al., 2017] Zhu, J.-Y., Park, T., Isola, P., and Efros, A. A. (2017). Unpaired image-to-image translation
> using cycle-consistent adversarial networks. In Proceedings of IEEE/CVF Computer Vision and Pattern
> Recognition (CVPR), pages 2223–2232.
>
> [Zhuang et al., 2020] Zhuang, F., Qi, Z., Duan, K., Xi, D., Zhu, Y., Zhu, H., Xiong, H., and He, Q. (2020). A
> comprehensive survey on transfer learning. Proceedings of the IEEE, 109(1):43–76.

---

### Meta-Review · Area_Chair_aFSE · 2023-12-18

**Metareview:**

This paper studies the unsupervised domain translation problem, and makes two main contributions in this regard: shedding light on an identifiability issue that arises in the presence of measure-preserving automorphisms, and proposing a method to mitigate it by matching distributions over variable-induced subsets of the data.

The paper elicited mixed reviews, with almost unanimous agreement on the quality of writing, the motivation of the approach, and the overall interest to the community of the problem studied. On the other hand, various reviewers raised concerns about the strength/justification of the assumptions underpinning the main theoretical results, and some important questions that were not answered in the original submissions.

In the reviewer-author discussion period, the authors clarified some misconceptions about the strength of the assumptions (e.g., any invertible map, instead of requiring those to be Optimal Transport maps). This seems to address the main concern of reviewer 4LhS, who nevertheless did not respond nor modify their score. Similarly, the weakness expressed by reviewer bENZ refer almost exclusively to typos or notational issues, which in my opinion should have prompted them to reconsider their score once the authors clarified/corrected those issues.

Overall, I believe this paper should ultimately be of interest to the machine learning community in so far as it provides novel and interesting theoretical insights about the nature of UDA problems.

**Justification For Why Not Higher Score:**

The novelty, strength, and applicability of the theoretical framework does not justify, in my opinion, a higher score .

**Justification For Why Not Lower Score:**

This paper should ultimately be of interest to the machine learning community in so far as it provides novel and interesting theoretical insights about the nature of UDA problems. Additionally, I believe the authors properly addressed most, if not all of the material concerns raised by the reviewers, leaving little support for a rejection decision.

---

### Decision · Program_Chairs · 2024-01-16

Accept (poster)